# A Penalty Approach for Differentiation Through Black-Box Quadratic Programming Solvers

Yuxuan Linghu [1]   Zhiyuan Liu [2]   Qi Deng [1]

## Abstract

Differentiating through the solution of a quadratic program (QP) is a central problem in differentiable optimization. Most existing approaches differentiate through the Karush–Kuhn–Tucker (KKT) system, but their computational cost and numerical robustness can degrade at scale. To address these limitations, we propose dXPP, a penalty-based differentiation framework that decouples QP solving from differentiation. In the solving step (forward pass), dXPP is solver-agnostic and can leverage any black-box QP solver. In the differentiation step (backward pass), we map the solution to a smooth approximate penalty problem and implicitly differentiate through it, requiring only the solution of a much smaller linear system in the primal variables. This approach bypasses the difficulties inherent in explicit KKT differentiation and significantly improves computational efficiency and robustness. We evaluate dXPP on various tasks, including randomly generated QPs, large-scale sparse projection problems, and a real-world multi-period portfolio optimization task. Empirical results demonstrate that dXPP is competitive with KKT-based differentiation methods and achieves substantial speedups on large-scale problems. Our implementation is open source and available at https://github.com/mmmmmmlinghu/dXPP.

## 1. Introduction

Differentiable optimization has emerged as a powerful paradigm for incorporating optimization problems into end-to-end learning pipelines. By embedding optimization layers that can backpropagate gradients, this approach enables

---

[1]Antai College of Economics and Management, Shanghai Jiao Tong University [2]The University of Chicago. Correspondence to: Qi Deng <qdeng24@sjtu.edu.cn>.

*Proceedings of the 43rd International Conference on Machine Learning*, Seoul, South Korea. PMLR 306, 2026. Copyright 2026 by the author(s).

automatic differentiation through complex decision-making processes, allowing model parameters to be learned from task-level objectives. The key advantage of differentiable optimization lies in its ability to directly encode optimization structure and enforce hard constraints (e.g., feasibility, optimality) that standard neural network layers cannot naturally guarantee (Amos & Kolter, 2017). This capability has proven valuable across diverse applications, including inventory control (Donti et al., 2017), portfolio optimization (Ye et al., 2020), and other data-driven decision-making tasks.

We consider the following convex quadratic program, parameterized by some $\theta \in \mathbb{R}^s$:

$$
\begin{aligned}
z^\star(\theta) = \arg\min_{z \in \mathbb{R}^n} \quad & \frac{1}{2} z^\top P(\theta) z + q(\theta)^\top z \\
\text{s.t.} \quad & A(\theta) z = b(\theta), \\
& C(\theta) z \leq d(\theta),
\end{aligned}
\tag{1}
$$

where $P(\theta) \in \mathbb{S}_{++}^n$ is symmetric positive definite, $q(\theta) \in \mathbb{R}^n$, $A(\theta) \in \mathbb{R}^{p \times n}$, $b(\theta) \in \mathbb{R}^p$, $C(\theta) \in \mathbb{R}^{m \times n}$, and $d(\theta) \in \mathbb{R}^m$ are differentiable mappings of $\theta$. We assume problem (1) is feasible for all $\theta$ of interest. Since the objective is strictly convex, the primal optimizer $z^\star(\theta)$ is unique. For notational simplicity, we omit the explicit dependence on $\theta$ when it is clear from context.

Our goal is to compute $\partial_\theta z^\star(\theta) \in \mathbb{R}^{n \times s}$, the sensitivity of the optimal solution with respect to the parameters. Early approaches often couple the layer with a customized solver to compute the gradient, which often leads to a slower and less scalable forward pass in practice (Amos & Kolter, 2017; Agrawal et al., 2019a). More recent solver-agnostic methods (e.g., BPQP and dQP) treat the forward pass as a black-box call to a mature optimizer (Pan et al., 2024; Magoon et al., 2025). Yet, their backward pass still necessitates solving a large linear system derived from the Karush–Kuhn–Tucker (KKT) equations, which can be computationally expensive for large-scale problems.

This paper focuses on differentiation through quadratic programs (QPs). While mature black-box solvers make the forward pass efficient, backpropagation remains a bottleneck: KKT-based differentiation requires solving a large saddle-

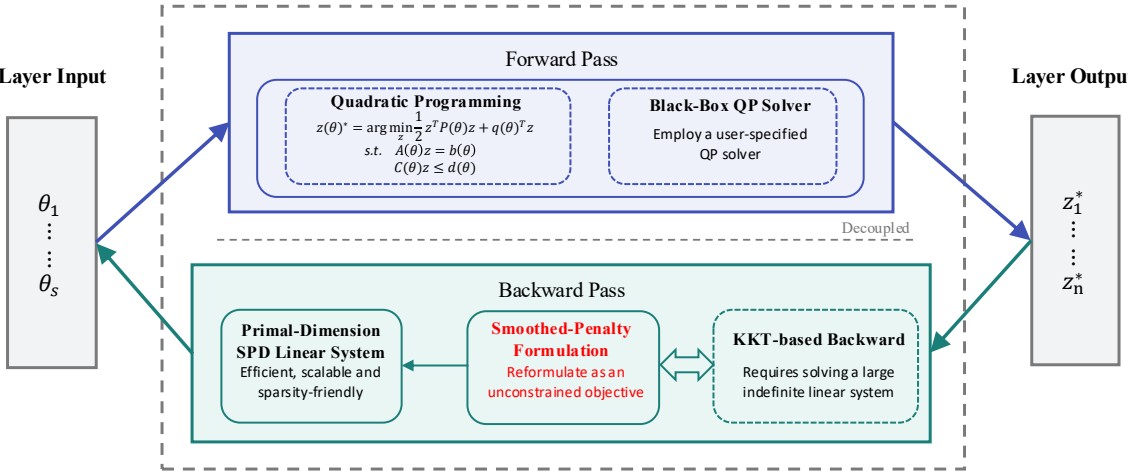

*Figure 1.* The learning workflow of dXPP: the previous layer outputs parameters $\theta$ and the QP layer returns the optimal solution $z^\star$ in the forward pass; the backward pass propagates loss gradients for end-to-end learning. By decoupling solving and differentiating through a penalty-based reformulation, dXPP enables efficient and scalable training with black-box QP solvers.

point linear system whose cost can scale cubically and becomes prohibitive for dense constraints or high-dimensional QPs. We propose dXPP, a new method for **d**ifferentiating through black-box conve**x** Q**P** solvers based on a **p**enalty formulation. Specifically, dXPP incorporates the constraints into the objective function and differentiates through the resulting unconstrained penalized problem. As shown in Figure 1, it serves as a plug-and-play layer that pairs any advanced QP solver (e.g., Gurobi) in the forward pass with a backward pass that reduces to solving symmetric positive definite (SPD) linear systems in primal variables, enabling an efficient, scalable, and sparsity-friendly gradient module.

Our contributions can be summarized as follows:

1. We present dXPP, a penalty-based approach to differentiating through black-box QP solvers that bypasses KKT differentiation and reduces the backward pass to a primal-dimensional SPD linear system, which is typically smaller and better conditioned than the KKT system, and remains well-defined under degeneracy due to smoothing.

2. We prove that the sensitivity computed from the smoothed penalty objective converges to the exact KKT sensitivity as the smoothing parameter tends to zero.

3. We evaluate dXPP on diverse benchmarks and a real-world portfolio optimization task. dXPP consistently outperforms KKT-based methods and achieves state-of-the-art performance for differentiating QPs.

## 2. Related Work

**Differentiable Optimization** Classical differentiable QP layers compute gradients by implicitly differentiating the KKT optimality conditions, as introduced in OptNet (Amos & Kolter, 2017). While effective at small- to medium-scale, KKT differentiation requires solving large indefinite linear systems in the backward pass and can become numerically fragile under active-set changes or degeneracy. To improve scalability, a parallel line of work differentiates through iterative-solver fixed points using residual-based implicit differentiation or Jacobian approximations (Sun et al., 2023; Butler & Kwon, 2023a; Butler, 2023), and augmented-Lagrangian variants further enhance robustness to infeasibility (Bambade et al., 2024). Beyond QPs, differentiable optimization has been extended to cone programs and disciplined convex programs (Agrawal et al., 2019b;a) and supported by modular implicit-differentiation toolkits (Blondel et al., 2022; Pineda et al., 2022), albeit often at the expense of QP-specific efficiency. More recently, work has revisited the backward mechanism itself: Pan et al. (2024) decouple the backward pass via simplified KKT structure, while Magoon et al. (2025) provide a solver-agnostic interface by characterizing the solution map through the active set.

**Penalty Methods** Classic work on exact penalties established that, for sufficiently large penalty parameters, local minimizers of the penalized objective coincide with those of the original constrained problem, with the required threshold governed by the optimal Lagrange multipliers (Evans et al., 1973; Han & Mangasarian, 1979; Di Pillo & Grippo, 1989; Bertsekas & Koksal, 2000). In parallel, Augmented Lagrangian Methods (ALM) combine penalty

terms with dual variables and remain a core paradigm for constrained nonlinear optimization (Hestenes, 1969; Powell, 1969; Rockafellar, 1974). A key practical issue is that widely used exact penalties are nonsmooth, motivating a broad literature on smoothing techniques that trade controlled approximation error for better numerical behavior (Nesterov, 2005; Meng et al., 2006; Xu et al., 2011). More recently, smooth exact-penalty constructions have been developed that preserve exactness while improving compatibility with gradient-based algorithms in general constrained settings (Dolgopolik, 2016; Xu et al., 2019; Estrin et al., 2020). These ideas also connect to modern constrained learning perspectives, including exact Augmented Lagrangian formulations and scalable large-scale solvers (Hong et al., 2023), as well as recent surveys of ALM and their advances (Deng et al., 2025).

## 3. Method

In this section, we develop dXPP, a framework for differentiating through convex quadratic programs. Our approach adopts a smoothed penalty reformulation to reduce the backward pass to solving a primal-scale SPD linear system. We begin by reviewing differentiation through the KKT system.

### 3.1. Differentiating Through the KKT Conditions

Since (1) is a strictly convex QP with affine constraints, its optimality is characterized by the Karush–Kuhn–Tucker (KKT) conditions. Introducing the Lagrange multipliers $\nu \in \mathbb{R}^p$ for the equality constraints and $\mu \in \mathbb{R}^m$ for the inequality constraints, the triplet $(z^\star, \nu^\star, \mu^\star)$ is primal-dual optimal if and only if it satisfies

$$
\begin{aligned}
& Pz^\star + q + A^\top \nu^\star + C^\top \mu^\star = 0, \\
& Az^\star = b, \qquad Cz^\star \leq d, \\
& \mu^\star \geq 0, \\
& \mu_i^\star (Cz^\star - d)_i = 0, \quad i = 1, \ldots, m.
\end{aligned} \tag{2}
$$

Traditional layers obtain $\partial_\theta z^\star$ by implicitly differentiating the KKT conditions. Let $U(z, \nu, \mu; \theta)$ collect the equations in (2) as $U(z, \nu, \mu; \theta) = \begin{bmatrix} Pz + q + A^\top \nu + C^\top \mu \\ Az - b \\ \mathrm{diag}(\mu)(Cz - d) \end{bmatrix}$.

Differentiating $U(z^\star, \nu^\star, \mu^\star; \theta) = 0$ with respect to $\theta$ yields the linear system

$$
\begin{bmatrix} P & A^\top & C^\top \\ A & 0 & 0 \\ \mathrm{diag}(\mu^\star)C & 0 & \mathrm{diag}(Cz^\star - d) \end{bmatrix} \begin{bmatrix} \partial_\theta z^\star \\ \partial_\theta \nu^\star \\ \partial_\theta \mu^\star \end{bmatrix}
$$
$$
= - \begin{bmatrix} (\partial_\theta P) z^\star + \partial_\theta q + (\partial_\theta A^\top) \nu^\star + (\partial_\theta C^\top) \mu^\star \\ (\partial_\theta A) z^\star - \partial_\theta b \\ \mathrm{diag}(\mu^\star)\big((\partial_\theta C) z^\star - \partial_\theta d\big) \end{bmatrix}. \tag{3}
$$

Under the linear independence constraint qualifica-

tion (LICQ) and strict complementarity, the linear system in (3) is nondegenerate, so the implicit system admits a unique solution for $\partial_\theta(z^\star, \nu^\star, \mu^\star)$. Degeneracy occurs precisely when there exists a weakly active inequality with $(Cz^\star - d)_i = 0$ but $\mu_i^\star = 0$ (Amos & Kolter, 2017). Define the active set $\mathcal{A} = \{i : (Cz^\star - d)_i = 0\}$ and the inactive set $\mathcal{I} = \{1, \ldots, m\} \setminus \mathcal{A}$. For any inactive constraint $i \in \mathcal{I}$, we have $(Cz^\star - d)_i < 0$ and hence $\mu_i^\star = 0$ and $\partial_\theta \mu_i^\star = 0$. Therefore, we can eliminate the inactive multipliers and retain only the active rows, yielding the reduced system

$$
\begin{bmatrix} P & A^\top & C_{\mathcal{A}}^\top \\ A & 0 & 0 \\ C_{\mathcal{A}} & 0 & 0 \end{bmatrix} \begin{bmatrix} \partial_\theta z^\star \\ \partial_\theta \nu^\star \\ \partial_\theta \mu_{\mathcal{A}}^\star \end{bmatrix}
$$
$$
= - \begin{bmatrix} (\partial_\theta P) z^\star + \partial_\theta q + (\partial_\theta A^\top) \nu^\star + (\partial_\theta C_{\mathcal{A}}^\top) \mu_{\mathcal{A}}^\star \\ (\partial_\theta A) z^\star - \partial_\theta b \\ (\partial_\theta C_{\mathcal{A}}) z^\star - \partial_\theta d_{\mathcal{A}} \end{bmatrix}, \tag{4}
$$

where $C_{\mathcal{A}}$, $d_{\mathcal{A}}$, and $\mu_{\mathcal{A}}^\star$ denote the restrictions of $C$, $d$, and $\mu^\star$ to the active inequalities. Here, we define $\partial_\theta z^\star$ computed in (4) as $Z_{\mathrm{KKT}}$.

### 3.2. Penalty Reformulation and Implicit Differentiation

For a fixed parameter $\theta$, let the objective function be $f(z) = \frac{1}{2} z^\top P z + q^\top z$. Following Nocedal & Wright (2006), we introduce the exact penalty objective:

$$
F(z; \theta, \rho, \alpha) = f(z) + \rho \|Az - b\|_1 + \alpha \|[Cz - d]_+\|_1, \tag{5}
$$

where $[u]_+ = \max\{u, 0\}$ is taken element-wise. The corresponding penalized optimization problem is

$$
\min_z F(z; \theta, \rho, \alpha). \tag{6}
$$

The necessary stationary condition for this penalty problem is given by:

$$
0 \in \nabla f(z) + \rho A^\top \partial \|Az - b\|_1 + \alpha C^\top \partial \|[Cz - d]_+\|_1. \tag{7}
$$

Specifically, at any $z$ for which $Az = b$, each component of $\partial \|Az - b\|_1$ belongs to $[-1, 1]$; and at any $z$ such that $Cz \leq d$, each component of $\partial \|[Cz - d]_+\|_1$ is zero if $(Cz - d)_i < 0$ and lies in $[0, 1]$ if $(Cz - d)_i = 0$. The following standard result shows that, for sufficiently large penalty weights, the exact penalty formulation is equivalent to the original QP. For completeness, the proof is given in Appendix A.

**Proposition 3.1.** *Let $(z^\star, \nu^\star, \mu^\star)$ be an optimal primal–dual solution of the QP (1). If the penalty weights satisfy*

$$
\rho \geq \|\nu^\star\|_\infty, \quad \alpha \geq \|\mu^\star\|_\infty, \tag{8}
$$

*then $z^\star$ is also a minimizer of the exact penalty problem (6).*

*Remark* 3.2. While the exact penalty problem may seem more tractable, recovering the solution to the original QP

generally requires careful selection of the penalty parameters, which in turn typically necessitates knowledge of the optimal primal–dual solution. Fortunately, modern QP solvers provide optimal dual variables as a byproduct at little additional computational cost, enabling the explicit construction of an equivalent penalty formulation. Importantly, in our framework, we do not directly solve the penalty problem; rather, we employ this reformulation solely to facilitate a more efficient implicit differentiation scheme.

The exact penalty objective is nonsmooth. To obtain stable derivatives through the penalty reformulation, we replace the hinge and $\ell_1$-norm terms in (5) with a softplus-smoothed approximation. Following Li et al. (2023), we define the softplus function

$$p_\delta(t) \triangleq \delta \log\big(1 + \exp(t/\delta)\big), \text{ where } \delta > 0.$$

It is easy to see that $p_\delta(\cdot)$ is convex and twice continuously differentiable. We rewrite the equality $\ell_1$-norm terms using $|u| = [u]_+ + [-u]_+$. We then apply softplus smoothing to the resulting hinge penalties for both equality and inequality constraints. Accordingly, we define the smoothed penalty objective

$$\Phi_\delta(z; \theta) = f(z) + \alpha \sum_{i=1}^{m} p_\delta\big((Cz - d)_i\big) \quad (9)$$

$$+ \rho \sum_{j=1}^{p} \Big(p_\delta\big((Az - b)_j\big) + p_\delta\big(-(Az - b)_j\big)\Big).$$

Let $z_\delta^\star(\theta) = \arg\min_z \Phi_\delta(z; \theta)$. We differentiate the stationarity condition $\nabla_z \Phi_\delta\big(z_\delta^\star; \theta\big) = 0$ with respect to $\theta$. By the implicit function theorem, we have

$$\partial_\theta z_\delta^\star = -\Big(\nabla_{zz}^2 \Phi_\delta\big(z_\delta^\star; \theta\big)\Big)^{-1} \nabla_{z\theta}^2 \Phi_\delta\big(z_\delta^\star; \theta\big). \quad (10)$$

### 3.3. Plug-in Sensitivity and Consistency Analysis

In practice, we do not compute the exact minimizer $z_\delta^\star$ of the smoothed penalty objective (9). Rather, we utilize the primal and dual solution obtained from the QP solver and substitute it into (10). We now assume strict complementarity, whereby the active set $\mathcal{A}$ remains locally invariant. Further, we assume that the inactive constraints are strictly separated; that is, there exists $\gamma > 0$ such that $(Cz^\star - d)_i \leq -\gamma$ for all $i \in \mathcal{I}$.

We set up some notation. First, define the stacked active constraint function $g(z, \theta) \triangleq \begin{bmatrix} Az - b \\ C_\mathcal{A} z - d_\mathcal{A} \end{bmatrix} \in \mathbb{R}^{p+|\mathcal{A}|}$ with its Jacobian given by $B \triangleq \nabla_z g(z, \theta) = \begin{bmatrix} A \\ C_\mathcal{A} \end{bmatrix} \in \mathbb{R}^{(p+|\mathcal{A}|) \times n}$. Let $g_\theta \triangleq \partial_\theta g(z^\star, \theta) \in \mathbb{R}^{(p+|\mathcal{A}|) \times s}$ and $G \triangleq \nabla_{z\theta}^2 f(z^\star; \theta) \in \mathbb{R}^{n \times s}$. We further define $W \triangleq \begin{bmatrix} \frac{\rho}{2} I_p & 0 \\ 0 & \frac{\alpha}{4} I_{|\mathcal{A}|} \end{bmatrix}$.

We begin by deriving the expression for $\nabla_{zz}^2 \Phi_\delta(\cdot; \theta)$ and subsequently evaluate it at $z^\star$. The resulting formula for $\nabla_{zz}^2 \Phi_\delta\big(z^\star; \theta\big)$ is presented in Proposition 3.3, with the proof detailed in Appendix B.

**Proposition 3.3.** *Let $C_\mathcal{I}$ and $d_\mathcal{I}$ denote the restriction of $(C, d)$ to inactive rows, and $s_\mathcal{I} = C_\mathcal{I} z^\star - d_\mathcal{I} \in \mathbb{R}^{|\mathcal{I}|}$. Define $E_\delta = \alpha C_\mathcal{I}^\top \text{diag}\big(p_\delta''(s_\mathcal{I})\big) C_\mathcal{I}$. Then*

$$\nabla_{zz}^2 \Phi_\delta\big(z^\star; \theta\big) = P + \tfrac{1}{\delta} B^\top W B + E_\delta. \quad (11)$$

*Moreover, $\lim_{\delta \to 0} \|E_\delta\|_2 = 0$.*

We now proceed to compute $\nabla_{z\theta}^2 \Phi_\delta\big(z_\delta^\star; \theta\big)$ in closed form. The proof is deferred to Appendix C.

**Proposition 3.4.** *With $\psi_\delta(t) = p_\delta(t) + p_\delta(-t)$ applied elementwise, we have*

$$
\begin{aligned}
&\nabla_{z\theta}^2 \Phi_\delta\big(z_\delta^\star; \theta\big) \\
&= \nabla_{z\theta}^2 f\big(z_\delta^\star; \theta\big) \\
&\quad + (\partial_\theta A)^\top \big(\rho\, \psi_\delta'(Az_\delta^\star - b)\big) \\
&\quad + (\partial_\theta C_\mathcal{A})^\top \big(\alpha\, p_\delta'(C_\mathcal{A} z_\delta^\star - d_\mathcal{A})\big) \\
&\quad + A^\top \text{Diag}\big(\rho\, \psi_\delta''(Az_\delta^\star - b)\big)(\partial_\theta A z_\delta^\star - \partial_\theta b) \\
&\quad + C_\mathcal{A}^\top \text{Diag}\big(\alpha\, p_\delta''(C_\mathcal{A} z_\delta^\star - d_\mathcal{A})\big)(\partial_\theta C_\mathcal{A} z_\delta^\star - \partial_\theta d_\mathcal{A}) \\
&\quad + (\partial_\theta C_\mathcal{I}^\top)\big(\alpha\, p_\delta'(C_\mathcal{I} z_\delta^\star - d_\mathcal{I})\big) \\
&\quad + C_\mathcal{I}^\top \text{Diag}\big(\alpha\, p_\delta''(C_\mathcal{I} z_\delta^\star - d_\mathcal{I})\big)\partial_\theta(C_\mathcal{I} z_\delta^\star - d_\mathcal{I}).
\end{aligned}
\quad (12)
$$

As previously discussed, the evaluation of $\nabla_{z\theta}^2 \Phi_\delta\big(z_\delta^\star; \theta\big)$ is performed at $z_\delta^\star$. In practice, we employ a plug-in approach that utilizes the solution obtained from the QP solver. First, consider the term

$$
\begin{aligned}
&(\partial_\theta A)^\top \big(\rho\, \psi_\delta'(Az_\delta^\star - b)\big) \\
&\quad + (\partial_\theta C_\mathcal{A})^\top \big(\alpha\, p_\delta'(C_\mathcal{A} z_\delta^\star - d_\mathcal{A})\big).
\end{aligned}
$$

where $\rho\, \psi_\delta'(Az_\delta^\star - b)$ and $\alpha\, p_\delta'(C_\mathcal{A} z_\delta^\star - d_\mathcal{A})$ act as proxies for the Lagrange multipliers. To achieve a more accurate representation, we replace this contribution with $(\partial_\theta B^\top)\, y^\star$, where $y^\star \in \mathbb{R}^{p+|\mathcal{A}|}$ denotes the actual stacked KKT multipliers associated with the active constraints.

Next, for the inactive constraints, define

$$
\begin{aligned}
F_\delta \triangleq &(\partial_\theta C_\mathcal{I}^\top)\big(\alpha\, p_\delta'(C_\mathcal{I} z_\delta^\star - d_\mathcal{I})\big) \\
&+ C_\mathcal{I}^\top \text{Diag}\big(\alpha\, p_\delta''(C_\mathcal{I} z_\delta^\star - d_\mathcal{I})\big)\, \partial_\theta(C_\mathcal{I} z_\delta^\star - d_\mathcal{I}).
\end{aligned}
$$

Since $\lim_{\delta \to 0} z_\delta^\star = z^\star$, for sufficiently small $\delta$ we have $(C_\mathcal{I} z_\delta^\star - d_\mathcal{I})_i \leq -\gamma + o(\gamma)$ for all $i \in \mathcal{I}$. Under the strict separation assumption for inactive constraints, $F_\delta$ exhibits the same exponential decay as $E_\delta$. Specifically, there exists $c_F > 0$ such that $\|F_\delta\|_2 \leq \frac{c_F}{\delta} e^{-(\gamma+o(\gamma))/\delta}$. Thus, although $F_\delta$ depends on $\delta$, it is asymptotically negligible.

**Algorithm 1** dXPP: Differentiation through Black-Box Quadratic Programming Solvers and Penalty Formulation

---

**Input:** parameters $\theta$ defining $P, q, A, b, C, d$; smoothing strength $\delta > 0$; penalty strength $\zeta \geq 1$
**Output:** $z^\star$ and Jacobian $\partial_\theta z^\star$

**Forward pass:**
Solve (1) using a black-box QP solver to obtain $(z^\star, \nu^\star, \mu^\star)$
Set $\rho = \zeta \|\nu^\star\|_\infty, \quad \alpha = \zeta \|\mu^\star\|_\infty$

**Backward pass:**
Form the smoothed penalty objective in (9)
Compute $\partial_\theta z^\star$ using (13)

---

For the remaining terms in (12), we replace $z_\delta^\star$ with $z^\star$. In particular, we take $\nabla_{z\theta}^2 f(z; \theta)$ and set $G = \nabla_{z\theta}^2 f(z^\star; \theta)$. Similarly, for the active curvature block, we substitute $z_\delta^\star$ with $z^\star$:

$$A^\top \text{Diag}\big(\rho\, \psi_\delta''(Az^\star - b)\big)(\partial_\theta Az^\star - \partial_\theta b)$$
$$+ C_\mathcal{A}^\top \text{Diag}\big(\alpha\, p_\delta''(C_\mathcal{A} z^\star - d_\mathcal{A})\big)(\partial_\theta C_\mathcal{A} z^\star - \partial_\theta d_\mathcal{A})$$
$$= \tfrac{1}{\delta} B^\top W g_\theta.$$

Bringing these components together, we obtain the plug-in estimate $Z_\delta^{\text{plug}}$ by solving the following linear system:

$$\begin{aligned}
\big(P + \tfrac{1}{\delta} B^\top W B + E_\delta\big) Z_\delta^{\text{plug}} \\
= -\big(G + (\partial_\theta B^\top) y^\star + \tfrac{1}{\delta} B^\top W g_\theta + F_\delta\big).
\end{aligned} \tag{13}$$

Empirically, we find that $Z_\delta^{\text{plug}}$ closely matches the reference gradients obtained by differentiating the KKT system of the original constrained problem (see Section 4.1). Next, we show that the plug-in sensitivity $Z_\delta^{\text{plug}}$ asymptotically converges to the exact gradient $Z_{\text{KKT}}$. The proof is provided in Appendix D.

**Theorem 3.5.** *Suppose $P$ is positive definite and that both LICQ and strict complementarity conditions are satisfied. Then, as $\delta \to 0$, the approximate sensitivity $Z_\delta^{\text{plug}}$ defined in* (13) *converges to the exact gradient $Z_{\text{KKT}}$ given by* (4).

### 3.4. Computational Advantages

The complete training framework of dXPP is outlined in Algorithm 1. This framework comprises solving the forward QP with a black-box solver, penalty parameter scaling, softplus-based smoothing, and the implicit backward differentiation step. We now proceed to discuss the computational advantages offered by dXPP.

**Solver-agnostic forward pass**   Similar to dQP (Magoon et al., 2025), the forward map of dXPP can be obtained by any convex QP solver, provided that the solver returns a primal solution $z^\star$ together with the corresponding dual multipliers $(\nu^\star, \mu^\star)$ for the equality and inequality constraints. These multipliers are used only to set the penalty magnitudes $(\rho, \alpha)$. This lets us flexibly choose the solver to match the underlying problem structure and hardware capabilities.

**dXPP as an embedding layer**   When dXPP is embedded in a deep learning model, it is unnecessary to construct the Jacobian $Z_\delta^{\text{plug}}$ explicitly. Let $\mathcal{L}$ denote the loss function, $r = \partial \mathcal{L}/\partial z^\star \in \mathbb{R}^n$ the upstream gradient, and $H = P + \frac{1}{\delta} B^\top W B + E_\delta$ the penalty Hessian. The vector-Jacobian product can then be computed by first solving $Hu = r$ for $u \in \mathbb{R}^n$, and subsequently evaluating $\partial_\theta \mathcal{L} = -\big(G + (\partial_\theta B^\top) y^\star + \frac{1}{\delta} B^\top W g_\theta + F_\delta\big)^\top u$. Accordingly, each backward pass entails solving a single $n \times n$ SPD linear system per evaluation, independent of the dimension $s$ of the problem parameters.

**Reduction to primal-dimensional linear systems**   Traditional approaches to differentiating through QP layers rely on differentiating the KKT system, which leads to solving an indefinite linear system of size $n + p + m$ as in (3), or $n+p+|\mathcal{A}|$ as in (4) after restricting to the active set $\mathcal{A}$ (Amos & Kolter, 2017; Agrawal et al., 2019a; Magoon et al., 2025). In contrast, the backward pass (13) in dXPP only involves the primal variable dimension $n$. Furthermore, the matrix $H$ inherits and integrates the sparsity structures of both $P$ and $B$. Owing to its symmetric positive definite (SPD) nature, $H$ is well suited for state-of-the-art sparse Cholesky factorizations (Chen et al., 2008) and preconditioned conjugate gradient methods (Nocedal & Wright, 2006), which typically offer superior speed and numerical stability compared to indefinite KKT solvers at similar sparsity levels. Furthermore, dXPP yields a stable surrogate sensitivity even when the KKT system (4) is degenerate: when the LICQ or strict complementarity conditions do not hold, $H$ stays strictly positive definite as long as $P \in \mathbb{S}_{++}^n$. Consequently, the gradient $Z_\delta^{\text{plug}}$ remains well defined.

**Support for active-set pruning**   Unlike dQP (Magoon et al., 2025), in which the backward pass involves only the active constraints, the backward pass of dXPP explicitly accounts for inactive constraints. This distinction arises because the softplus smoothing term remains nonzero for $t < 0$. Nevertheless, the contributions from strictly inactive inequalities, represented by $E_\delta$ and $F_\delta$ in (13), vanish as $\delta \to 0$. In practice, we implement active-set pruning by omitting the $E_\delta$ and $F_\delta$ terms, and we observe empirically that this does not adversely affect performance.

## 4. Experiments

We evaluate dXPP by studying gradient accuracy on random strictly convex QPs, scalability on large-scale sparse pro-

*Table 1.* Gradient discrepancy between our softplus-penalty QP layer and dQP (Magoon et al., 2025). We report the mean and standard deviation of the relative difference. The QP size $n \times m$ denotes $n$ variables with $m$ equality and $m$ inequality constraints.

| QP size | Avg. $\epsilon_{\mathrm{rel}}$ | Std. $\epsilon_{\mathrm{rel}}$ |
|---|---|---|
| $10 \times 5$ | $1.91 \times 10^{-7}$ | $1.40 \times 10^{-7}$ |
| $50 \times 10$ | $8.55 \times 10^{-8}$ | $3.57 \times 10^{-8}$ |
| $100 \times 20$ | $2.64 \times 10^{-7}$ | $9.50 \times 10^{-8}$ |
| $500 \times 100$ | $5.21 \times 10^{-6}$ | $6.50 \times 10^{-7}$ |
| $1000 \times 200$ | $1.74 \times 10^{-5}$ | $2.25 \times 10^{-6}$ |
| $1500 \times 500$ | $5.36 \times 10^{-5}$ | $3.51 \times 10^{-6}$ |
| $3000 \times 1000$ | $1.91 \times 10^{-4}$ | $1.23 \times 10^{-5}$ |
| $5000 \times 2000$ | $5.14 \times 10^{-4}$ | $2.39 \times 10^{-5}$ |

jection problems and an end-to-end multi-period portfolio optimization task. We use the unified setting in Algorithm 1 with smoothing strength $\delta = 10^{-6}$ and penalty strength $\zeta = 10$. For a fair comparison, the forward QP solves for both dXPP and dQP are performed using Gurobi (Gurobi Optimization, LLC, 2024). Additional experimental details are provided in Appendix E.

### 4.1. Gradient Accuracy

We first assess whether the gradients produced by dXPP are numerically reliable. Following the benchmark of BPQP (Pan et al., 2024), we construct random strictly convex QPs of different sizes. The problem size is denoted by $n \times m$, where $n$ is the number of decision variables, and we use $m$ equality constraints and $m$ inequality constraints. Concretely, we sample a random matrix $P' \in \mathbb{R}^{n \times n}$ and construct a positive definite quadratic term $P = P'P'^{\top} + 10^{-6}I$ to ensure strict convexity. Linear terms and constraints are drawn from a standard normal distribution. To guarantee feasibility, we fix a reference point $z_0 = \mathbf{1}$ at the all-ones vector and set the constraint right-hand sides as $d = Cz_0 + \mathbf{1}$ and $b = Az_0$.

As a reference for gradient correctness, we compare our gradients against those produced by dQP (Magoon et al., 2025). Let $g_{\mathrm{dXPP}}$ be the gradient returned by our layer and $g_{\mathrm{dQP}}$ be the gradient returned by dQP. We report the relative difference

$$\epsilon_{\mathrm{rel}} = \frac{\|g_{\mathrm{dXPP}} - g_{\mathrm{dQP}}\|_2}{\|g_{\mathrm{dQP}}\|_2}.$$

Table 1 summarizes the mean and standard deviation of $\epsilon_{\mathrm{rel}}$ over multiple random instances for each problem size. For each size, we report statistics over 50 independently generated random instances. Across all tested problem sizes, the relative gradient discrepancy remains small. Specifically, the average $\epsilon_{\mathrm{rel}}$ is on the order of $10^{-7}$–$10^{-4}$ as the problem scale increases from $10 \times 5$ to $5000 \times 2000$. As expected, the discrepancy increases with the problem dimension and

the number of constraints, reflecting the accumulation of numerical errors in larger-scale instances. Nevertheless, even at the largest scale considered, the relative difference remains below $10^{-3}$, indicating that the backward pass of dXPP remains numerically reliable.

We further study the sensitivity of dXPP to the smoothing parameter $\delta$. To compare instances of different scales, we choose $\delta$ by a scale-aware rule $\delta \propto \rho_\delta \|\mathrm{KKT}\|$, rounded to the nearest power of ten, where $\rho_\delta$ is a multiplicative scale factor. Table 2 shows that the best accuracy is attained at an intermediate smoothing level: large $\delta$ introduces smoothing bias, while very small $\delta$ makes the backward system more numerically sensitive.

### 4.2. Scalability on Large-Scale Sparse Problems

We evaluate the scalability of dXPP on two large-scale sparse projection problems (Magoon et al., 2025). The first task is projection onto the probability simplex:

$$\arg\min_{z \in \mathbb{R}^n} \|x - z\|_2^2$$
$$\text{s.t.} \quad 0 \le z \le 1, \quad \sum_{i=1}^{n} z_i = 1, \qquad (P_1)$$

where the problem size corresponds to the dimension of $x$ and $z$. The second task is projection onto a chain with $\ell_\infty$-bounded differences:

$$\arg\min_{z \in \mathbb{R}^n} \sum_{j=1}^{n} \|x_j - z_j\|_2^2$$
$$\text{s.t.} \quad \|z_j - z_{j+1}\|_\infty \le 1, \ 1 \le j \le n-1, \qquad (P_2)$$

where the problem size corresponds to the chain length $n$. Both problems are strictly convex projections with simple, highly structured constraints, making them well-suited for evaluating differentiation scalability. We compare dXPP against dQP, OptNet, SCQPTH, and CVXPYLayers over a wide range of problem sizes and report wall-clock runtime in milliseconds. The results are summarized in Table 3 and Table 4.

We first note that dXPP and dQP share a key advantage in the forward pass: both are solver-agnostic. In particular, both leverage Gurobi, a commercial solver based on interior-point methods, which efficiently handles the forward solve even at the largest scale of $10^6$ variables. In contrast, OptNet relies on a custom primal–dual interior-point method implemented in PyTorch, while SCQPTH uses an ADMM-based operator-splitting solver implemented as a custom PyTorch module. Meanwhile, CVXPYLayers transforms the QP into a standard second-order cone program; at our scale, this reduction and the associated data-construction overhead become prohibitive. Consequently, these prior methods are limited to substantially smaller problem sizes,

*Table 2.* Effect of the smoothing scale on gradient accuracy. We choose $\delta \propto \rho_\delta \|\text{KKT}\|$ and report the mean and standard deviation of the relative gradient discrepancy.

| QP size | $\|\text{KKT}\|$ | $\rho_\delta = 10^{-5}$ | $\rho_\delta = 10^{-6}$ | $\rho_\delta = 10^{-7}$ | $\rho_\delta = 10^{-8}$ | $\rho_\delta = 10^{-9}$ |
|---|---|---|---|---|---|---|
| $10 \times 5$ | $4.92 \times 10^1$ | $3.67{\times}10^{-5}$ $\pm3.38{\times}10^{-5}$ | $1.48{\times}10^{-6}$ $\pm7.51{\times}10^{-7}$ | $\mathbf{1.91{\times}10^{-7}}$ $\mathbf{\pm1.40{\times}10^{-7}}$ | $9.07{\times}10^{-7}$ $\pm6.54{\times}10^{-7}$ | $1.11{\times}10^{-5}$ $\pm7.42{\times}10^{-6}$ |
| $100 \times 20$ | $1.43 \times 10^3$ | $1.07{\times}10^{-6}$ $\pm2.45{\times}10^{-7}$ | $2.03{\times}10^{-7}$ $\pm6.00{\times}10^{-8}$ | $\mathbf{1.03{\times}10^{-7}}$ $\mathbf{\pm1.60{\times}10^{-8}}$ | $2.61{\times}10^{-7}$ $\pm1.03{\times}10^{-7}$ | $3.02{\times}10^{-6}$ $\pm1.94{\times}10^{-6}$ |
| $1000 \times 200$ | $4.47 \times 10^4$ | $8.83{\times}10^{-7}$ $\pm1.70{\times}10^{-7}$ | $2.78{\times}10^{-7}$ $\pm5.10{\times}10^{-8}$ | $\mathbf{1.83{\times}10^{-7}}$ $\mathbf{\pm1.19{\times}10^{-8}}$ | $1.58{\times}10^{-6}$ $\pm2.03{\times}10^{-7}$ | $1.12{\times}10^{-5}$ $\pm1.37{\times}10^{-5}$ |
| $5000 \times 2000$ | $5.00 \times 10^5$ | $3.18{\times}10^{-6}$ $\pm2.40{\times}10^{-7}$ | $7.25{\times}10^{-7}$ $\pm6.10{\times}10^{-8}$ | $\mathbf{2.62{\times}10^{-7}}$ $\mathbf{\pm4.56{\times}10^{-9}}$ | $2.11{\times}10^{-6}$ $\pm1.52{\times}10^{-7}$ | $1.63{\times}10^{-5}$ $\pm1.21{\times}10^{-5}$ |

*Table 3.* Runtime analysis for projection onto the probability simplex.

| Solver | Metric | Problem Size | | | | | | | |
|---|---|---|---|---|---|---|---|---|---|
| | | 20 | 100 | 450 | 1000 | 4600 | 10000 | 100000 | 1000000 |
| dXPP | Backward (ms) | 0.72 | 0.82 | **0.91** | **0.92** | **1.09** | **1.51** | 8.35 | 226.21 |
| | Total (ms) | 5.01 | **5.13** | 6.27 | 7.73 | 22.48 | 46.09 | 494.02 | 5212.38 |
| dQP | Backward (ms) | 0.82 | 0.89 | 0.99 | 1.51 | 4.63 | 10.01 | 83.79 | 950.64 |
| | Total (ms) | 5.55 | 5.95 | 7.11 | 9.22 | 26.55 | 56.50 | 572.64 | 6103.44 |
| OptNet | Backward (ms) | **0.54** | **0.74** | 2.84 | 18.41 | 372.73 | 2647.40 | – | – |
| | Total (ms) | 5.47 | 8.21 | 46.48 | 286.28 | 4716.01 | 34121.95 | – | – |
| SCQPTH | Backward (ms) | 0.87 | 2.62 | – | – | – | – | – | – |
| | Total (ms) | 53.40 | 481.95 | – | – | – | – | – | – |
| CVXPYLayers | Backward (ms) | 1.23 | 1.72 | – | – | – | – | – | – |
| | Total (ms) | **3.15** | 7.37 | – | – | – | – | – | – |

with SCQPTH and CVXPYLayers failing to complete beyond dimensions of $10^2$–$10^3$.

In the backward pass, dXPP demonstrates superior scalability compared to dQP. For problem $(P_1)$, dXPP is competitive at small dimensions and becomes increasingly favorable as the dimension grows. Specifically, dXPP takes 226.21 ms at size $10^6$ versus 950.64 ms for dQP, yielding a $4.2\times$ speedup at the largest scale. For problem $(P_2)$, dXPP substantially outperforms dQP at large scales: at size $10^5$, the backward pass is 25.09 ms for dXPP versus 195.27 ms for dQP ($7.8\times$ speedup), and at size $10^6$ it is 333.42 ms versus 3071.18 ms ($9.2\times$ speedup). Overall, these results indicate that dXPP sustains stable and efficient differentiation over six orders of magnitude in the problem dimension, whereas the runtimes of other baseline layers grow rapidly and do not extend to the largest sizes.

To illustrate the scalability of dXPP compared with dQP, we report the backward runtime ratio $(t_{\text{dXPP}}^{\text{bwd}}/t_{\text{dQP}}^{\text{bwd}})$ in Figure 2. It can be seen that dXPP's backward time is consistently faster than dQP's backward time and exhibits significant

speedups as the problem size increases.

### 4.3. End-to-End Multi-Period Portfolio Optimization

We further demonstrate the scalability of dXPP on an end-to-end learning problem that couples prediction with multi-period portfolio optimization. We embed a multi-period mean-variance program (Boyd et al., 2017) as a differentiable layer parameterized by the predictor's return forecasts, and train the predictor by minimizing a portfolio-level decision loss computed from realized performance (Elmachtoub & Grigas, 2022). This setting is a typical application of decision-focused learning in finance, but it becomes challenging at scale (Butler & Kwon, 2023b).

In portfolio optimization, many asset weights commonly lie on their bounds (Sharpe, 1963), resulting in a large number of active constraints for which strict complementarity can fail (Nocedal & Wright, 2006). In such settings, KKT-based differentiation can yield severely ill-conditioned or even degenerate linear systems. As a result, sparse solvers may fail (Cannataro et al., 2016). Practical implementations

*Table 4.* Runtime analysis for projection onto chains.

| Solver | Metric | Problem Size | | | | | | | |
|---|---|---|---|---|---|---|---|---|---|
| | | 200 | 500 | 1000 | 2000 | 4000 | 10000 | 100000 | 1000000 |
| dXPP | Backward (ms) | **1.06** | **1.74** | **2.59** | **3.05** | **4.08** | **7.01** | **25.09** | **333.42** |
| | Total (ms) | **13.24** | **15.61** | **27.82** | **45.56** | **81.93** | **214.52** | **2495.59** | **29306.28** |
| dQP | Backward (ms) | 1.49 | 2.17 | 3.72 | 3.76 | 7.90 | 17.07 | 195.27 | 3071.18 |
| | Total (ms) | 14.14 | 16.97 | 29.30 | 47.08 | 85.00 | 225.94 | 2695.72 | 32201.43 |
| OptNet | Backward (ms) | 7.25 | 14.00 | 43.75 | 202.62 | 1190.40 | 12873.17 | – | – |
| | Total (ms) | 46.76 | 225.79 | 759.01 | 3210.28 | 19107.02 | 235462.01 | – | – |
| SCQPTH | Backward (ms) | 18.93 | 35.43 | 162.59 | 842.21 | 5002.86 | 68260.63 | – | – |
| | Total (ms) | 47.68 | 87.53 | 897.21 | 3512.24 | 14956.00 | 110042.63 | – | – |
| CVXPYLayers | Backward (ms) | 41.01 | – | – | – | – | – | – | – |
| | Total (ms) | 105.88 | – | – | – | – | – | – | – |

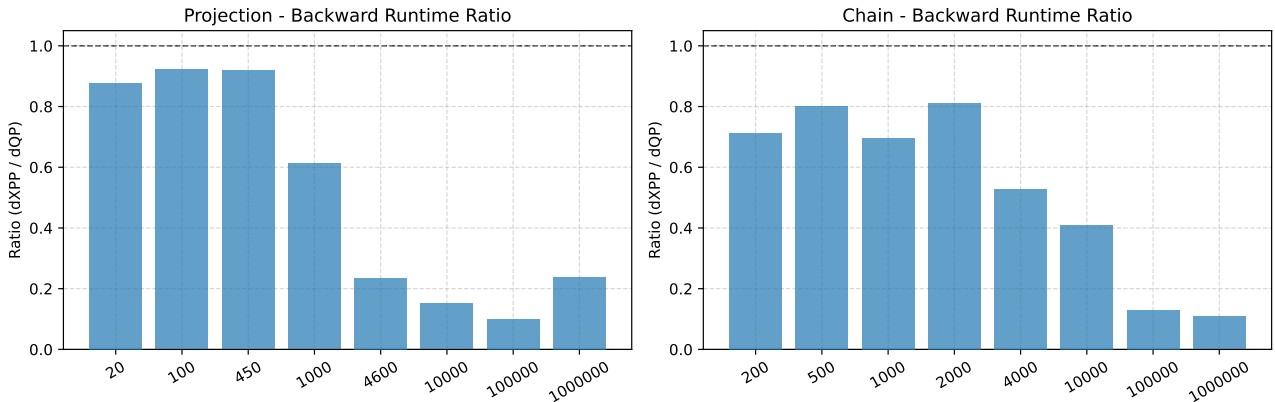

*Figure 2.* Backward runtime ratio ($t_{\text{dXPP}}^{\text{bwd}}/t_{\text{dQP}}^{\text{bwd}}$) for structured projection problems. The black dashed line represents the baseline (dQP's backward time normalized to 1). Exact results are reported in Table 3 and Table 4.

(e.g., Magoon et al. (2025)) often rely on damping or least-squares solves, which may introduce bias or instability in the resulting sensitivities. Dense factorizations with pivoting can improve numerical stability when solving these linear systems (Bunch & Kaufman, 1977). Accordingly, we run both dXPP and dQP in this dense mode to ensure numerical stability and a fair comparison.

We consider seven tradable ETFs (VTI, IWM, AGG, LQD, MUB, DBC, GLD) and use different investment horizons $H$, which directly control the scale of the optimization problem. At each decision date $t$, we solve a multi-stage mean-variance planning problem using predicted returns and execute the first-step allocation $w_{t+1}^\star$. The decision variables are the per-period portfolio weights $\{w_s\}_{s=t+1}^{t+H}$. This yields

the following bilevel formulation:

$$\min_\theta \sum_{s=t+1}^{t+H} \left( -r_s^\top w_s^\star(\theta) + \frac{\lambda}{2}(w_s^\star(\theta))^\top \Sigma_s w_s^\star(\theta) \right)$$

$$\text{s.t. } \mathbf{w}^\star(\theta) = \arg\min_{\mathbf{w}} \sum_{s=t+1}^{t+H} \left( -\hat{r}_s(\theta)^\top w_s + \frac{\lambda}{2} w_s^\top \hat{\Sigma}_s w_s \right)$$

$$\text{s.t. } w_s \geq 0, \quad \mathbf{1}^\top w_s = 1,$$

$$\|w_s - w_{s-1}\|_1 \leq \tau. \tag{14}$$

Here, $\lambda > 0$ is the risk aversion parameter, $\tau > 0$ is a global turnover budget, $\hat{r}_s(\theta)$ is produced by a linear predictor from past ETF returns, and $r_s$ denotes the realized return vector. For each period $s$, $\hat{\Sigma}_s$ denotes the covariance matrix estimated by returns before date $t$, whereas $\Sigma_s$ denotes the corresponding realized return covariance used to evaluate risk in the outer objective. Details of the training process are provided in Appendix E.2.

Table 5 and Figure 3 report average per-epoch runtime statistics as the investment horizon $H$ increases. We observe that

*Table 5.* Average per-epoch runtime for end-to-end multi-period portfolio optimization. Both dXPP and dQP are run in dense mode.

| Solver | Metric | Horizon $H$ | | | | | |
|--------|--------|------|------|------|------|------|------|
| | | 10 | 20 | 50 | 100 | 150 | 200 |
| dXPP | Backward (ms) | **1.71** | **2.91** | **10.29** | **36.66** | **68.72** | **113.97** |
| | Total (ms) | **11.91** | **18.07** | **60.58** | **210.41** | **496.52** | **977.98** |
| dQP | Backward (ms) | 17.68 | 68.00 | 694.68 | 3830.83 | 16583.49 | 39105.75 |
| | Total (ms) | 27.50 | 85.57 | 763.92 | 4049.86 | 17122.70 | 40066.11 |
| OptNet | Backward (ms) | 3.24 | 8.51 | 34.82 | 117.34 | 293.70 | 605.89 |
| | Total (ms) | 35.32 | 98.33 | 538.97 | 1911.39 | 4698.60 | 10978.19 |
| SCQPTH | Backward (ms) | 6.55 | 13.80 | 96.02 | 429.49 | 1777.38 | 3368.87 |
| | Total (ms) | 1074.22 | 1641.08 | 8121.46 | 41659.76 | 114059.42 | 194166.95 |
| CVXPYLayers | Backward (ms) | 7.76 | – | – | – | – | – |
| | Total (ms) | 25.52 | – | – | – | – | – |

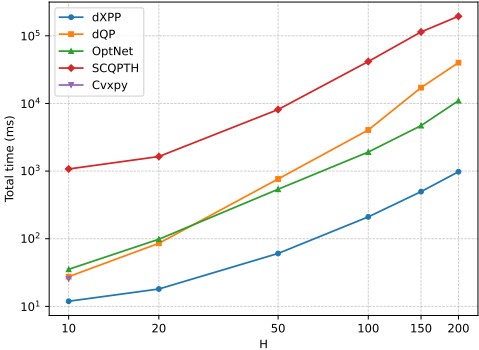

*Figure 3.* Visualization of the runtime results in Table 5.

dXPP is highly efficient and scales nearly linearly in backward time, increasing from 1.71 ms at $H{=}10$ to 113.97 ms at $H{=}200$. In contrast, dQP's backward runtime grows dramatically, reaching 39105.75 ms at $H{=}200$ (nearly $343\times$ slower than dXPP). OptNet scales more stably than dQP, but remains $5.3\times$ slower than dXPP in the backward pass at $H{=}200$ and results in higher overall training time. SCQPTH and CVXPYLayers either do not scale to larger horizons or hit memory limits. Moreover, dXPP achieves a faster forward pass due to the efficiency of the underlying Gurobi solver. Although both dXPP and dQP use Gurobi, dQP incurs substantial overhead when forming the KKT-based linear system (4) in dense mode. Overall, these results indicate that dXPP provides order-of-magnitude speedups while retaining the numerical robustness required for stable end-to-end learning in complex optimization settings where strict complementarity frequently fails.

## 5. Conclusion

We present dXPP, a penalty-based framework for differentiating through convex quadratic programs (QPs). Leveraging a softplus-smoothed penalty reformulation, dXPP decouples the forward and backward passes, reducing gradient computation to the solution of primal-dimensional linear systems, which circumvents the numerical challenges associated with KKT-based differentiation. Empirically, dXPP achieves significant speedups and maintains high accuracy across both random QPs and large-scale sparse projection problems, offering a solver-agnostic and scalable optimization layer for differentiable programming. While we have focused on convex QPs, the penalty-based differentiation framework of dXPP extends naturally to more general convex programs. Exploring such extensions to broader convex programming problems is a promising direction for future work.

## Acknowledgements

This project was supported in part by the National Natural Science Foundation of China (Grants 12571325, 72394364/72394360) and the Natural Science Foundation of Shanghai (Grant 24ZR1421300).

## Impact Statement

This work aims to advance the field of machine learning by improving differentiable optimization techniques. While our methods have the potential for broad societal impact, we do not identify any specific societal risks or concerns that require special attention at this time.

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

## A. Proof of Proposition 3.1

*Proof.* The function $F(\cdot; \theta, \rho, \alpha)$ is convex because both $f$ and the penalty terms are convex functions. Therefore, it is sufficient to show that $0 \in \partial F(z^\star)$. Without loss of generality, we assume that $\nu^\star$ and $\mu^\star$ are nonzero. If either were zero, the constraints would be trivially satisfied at $z^\dagger = \arg\min_z f(z)$, in which case, $z^\dagger = z^\star$, and for any $\rho, \alpha \geq 0$, $z^\dagger$ would clearly minimize $F(z; \theta, \rho, \alpha)$.

We first characterize suitable subgradients for the penalty terms at $z^\star$. Since $Az^\star = b$, we have

$$\partial \|Az^\star - b\|_1 = \{\, s \in \mathbb{R}^p : \ s_j \in [-1, 1], \ 1 \leq j \leq p \,\}.$$

Because $\rho \geq \|\nu^\star\|_\infty$, the vector $s := \nu^\star/\rho$ satisfies $s \in [-1, 1]^p$ and thus $A^\top \nu^\star = \rho A^\top s$.

Next, define $\phi(u) = \|[u]_+\|_1 = \sum_{i=1}^m \max\{u_i, 0\}$. Then

$$\partial \phi(u)_i = \begin{cases} \{0\}, & u_i < 0, \\ [0, 1], & u_i = 0, \\ \{1\}, & u_i > 0. \end{cases}$$

At the feasible point $z^\star$ we have $Cz^\star - d \leq 0$. By complementary slackness, $\mu_i^\star = 0$ whenever $(Cz^\star - d)_i < 0$. For indices with $(Cz^\star - d)_i = 0$, the assumption $\alpha \geq \|\mu^\star\|_\infty$ implies $\mu_i^\star/\alpha \in [0, 1]$. Therefore, if we define $t \in \mathbb{R}^m$ componentwise by

$$t_i := \begin{cases} 0, & (Cz^\star - d)_i < 0, \\ \mu_i^\star/\alpha, & (Cz^\star - d)_i = 0, \end{cases}$$

then $t \in \partial \|[Cz^\star - d]_+\|_1$ and $\alpha C^\top t = C^\top \mu^\star$.

Combining these choices with the KKT stationarity condition gives

$$0 = \nabla f(z^\star) + A^\top \nu^\star + C^\top \mu^\star = \nabla f(z^\star) + \rho A^\top s + \alpha C^\top t \in \nabla f(z^\star) + \rho A^\top \partial \|Az^\star - b\|_1 + \alpha C^\top \partial \|[Cz^\star - d]_+\|_1 = \partial F(z^\star).$$

Thus $0 \in \partial F(z^\star)$, and convexity of $F$ implies that $z^\star$ is a global minimizer of $\min_z F(z; \theta, \rho, \alpha)$.

$\square$

## B. Proof of Proposition 3.3

*Proof.* Let $\sigma(u) = \frac{1}{1 + e^{-u}}$. Then we have the following useful identities:

$$\begin{aligned}
p_\delta'(t) &= \sigma(t/\delta), \\
p_\delta''(t) &= \frac{1}{\delta} \sigma(t/\delta)\big(1 - \sigma(t/\delta)\big), \\
\psi_\delta'(t) &= \sigma(t/\delta) - \sigma(-t/\delta) = \tanh\big(t/(2\delta)\big), \\
\psi_\delta''(t) &= \frac{1}{2\delta} \operatorname{sech}^2\big(t/(2\delta)\big).
\end{aligned} \tag{15}$$

Hence $p_\delta''(0) = \frac{1}{4\delta}$ and $\psi_\delta''(0) = \frac{1}{2\delta}$. Moreover, if $t \leq -\gamma$, then $p_\delta'(t) \leq e^{-\gamma/\delta}$ and $p_\delta''(t) \leq \frac{1}{\delta} e^{-\gamma/\delta}$.

Differentiating $\Phi_\delta$ twice with respect to $z$, the objective contributes $P = \nabla_{zz}^2 f(z^\star; \theta)$. For the equality penalties, using $Az^\star - b = 0$ and $\psi_\delta''(0) = \frac{1}{2\delta}$, we have

$$\nabla_{zz}^2 \Big(\rho \sum_{j=1}^p \psi_\delta((Az - b)_j)\Big)\Big|_{z=z^\star} = \rho A^\top \operatorname{Diag}\big(\psi_\delta''(0)\big) A = \frac{\rho}{2\delta} A^\top A.$$

For active inequalities, using $C_\mathcal{A} z^\star - d_\mathcal{A} = 0$ and $p_\delta''(0) = \frac{1}{4\delta}$, we have

$$\nabla_{zz}^2 \Big(\alpha \sum_{i \in \mathcal{A}} p_\delta((Cz - d)_i)\Big)\Big|_{z=z^\star} = \alpha C_\mathcal{A}^\top \operatorname{Diag}\big(p_\delta''(0)\big) C_\mathcal{A} = \frac{\alpha}{4\delta} C_\mathcal{A}^\top C_\mathcal{A}.$$

For inactive constraints $i \in \mathcal{I}$, we have $(Cz^\star - d)_i \leq -\gamma$. The inactive Hessian component is

$$E_\delta = \alpha\, C_{\mathcal{I}}^\top \operatorname{Diag}\!\big(p_\delta''(s_{\mathcal{I}})\big)\, C_{\mathcal{I}}.$$

Since $p_\delta''(t) \leq \frac{1}{\delta} e^{-\gamma/\delta}$ for all $t \leq -\gamma$, it follows that $\|E_\delta\|_2 \leq \frac{c_E}{\delta} e^{-\gamma/\delta}$ for a constant $c_E > 0$ independent of $\delta$. Thus, we obtain the decomposition

$$\nabla_{zz}^2 \Phi_\delta(z^\star; \theta) = P + \frac{1}{\delta} B^\top W B + E_\delta. \tag{16}$$

$\square$

## C. Proof of Proposition 3.4

*Proof.* Differentiating (9) with respect to $z$, we obtain

$$\nabla_z \Phi_\delta(z; \theta) = \nabla_z f(z; \theta) + \rho A^\top \psi_\delta'(Az - b) + \alpha C^\top p_\delta'(Cz - d). \tag{17}$$

Differentiating (17) with respect to $\theta$ and evaluate at $z = z_\delta^\star$, we have

$$\begin{aligned}
\nabla_{z\theta}^2 \Phi_\delta\big(z_\delta^\star; \theta\big) = {}& \nabla_{z\theta}^2 f\big(z_\delta^\star; \theta\big) \\
&+ (\partial_\theta A^\top)\big(\rho\, \psi_\delta'(Az_\delta^\star - b)\big) + A^\top \operatorname{Diag}\big(\rho\, \psi_\delta''(Az_\delta^\star - b)\big)\, \partial_\theta(Az_\delta^\star - b) \\
&+ (\partial_\theta C^\top)\big(\alpha\, p_\delta'(Cz_\delta^\star - d)\big) + C^\top \operatorname{Diag}\big(\alpha\, p_\delta''(Cz_\delta^\star - d)\big)\, \partial_\theta(Cz_\delta^\star - d).
\end{aligned} \tag{18}$$

Next, we partition $C$ and $d$ by active and inactive sets:

$$C = \begin{bmatrix} C_{\mathcal{A}} \\ C_{\mathcal{I}} \end{bmatrix}, \qquad d = \begin{bmatrix} d_{\mathcal{A}} \\ d_{\mathcal{I}} \end{bmatrix}.$$

Then the inequality terms in (18) split as

$$(\partial_\theta C^\top)\big(\alpha\, p_\delta'(Cz_\delta^\star - d)\big) = (\partial_\theta C_{\mathcal{A}}^\top)\big(\alpha\, p_\delta'(C_{\mathcal{A}} z_\delta^\star - d_{\mathcal{A}})\big) + (\partial_\theta C_{\mathcal{I}}^\top)\big(\alpha\, p_\delta'(C_{\mathcal{I}} z_\delta^\star - d_{\mathcal{I}})\big),$$

$$\begin{aligned}
& C^\top \operatorname{Diag}\big(\alpha\, p_\delta''(Cz_\delta^\star - d)\big)\, \partial_\theta(Cz_\delta^\star - d) \\
&= C_{\mathcal{A}}^\top \operatorname{Diag}\big(\alpha\, p_\delta''(C_{\mathcal{A}} z_\delta^\star - d_{\mathcal{A}})\big)\, \partial_\theta(C_{\mathcal{A}} z_\delta^\star - d_{\mathcal{A}}) + C_{\mathcal{I}}^\top \operatorname{Diag}\big(\alpha\, p_\delta''(C_{\mathcal{I}} z_\delta^\star - d_{\mathcal{I}})\big)\, \partial_\theta(C_{\mathcal{I}} z_\delta^\star - d_{\mathcal{I}}).
\end{aligned}$$

Substituting it back into (18) yields (12). $\square$

## D. Proof of Theorem 3.5

*Proof.* By the notations defined in Section 3.3, (4) is equal to the following linear system:

$$\begin{bmatrix} P & B^\top \\ B & 0 \end{bmatrix} \begin{bmatrix} Z_{\text{KKT}} \\ \Lambda \end{bmatrix} = - \begin{bmatrix} G + (\partial_\theta B^\top) y^\star \\ g_\theta \end{bmatrix}, \tag{19}$$

where $\Lambda = \partial_\theta y^\star$. Next, for $Z_\delta^{\text{plug}}$, we define an auxiliary variable $\eta_\delta^{\text{plug}} = \frac{1}{\delta} W(B Z_\delta^{\text{plug}} + g_\theta) \in \mathbb{R}^{(p+|\mathcal{A}|) \times s}$. Then (13) is equivalent to the block system

$$\begin{bmatrix} P + E_\delta & B^\top \\ B & -\delta W^{-1} \end{bmatrix} \begin{bmatrix} Z_\delta^{\text{plug}} \\ \eta_\delta^{\text{plug}} \end{bmatrix} = - \begin{bmatrix} G + (\partial_\theta B^\top) y^\star + F_\delta \\ g_\theta \end{bmatrix}. \tag{20}$$

Let

$$K_0 = \begin{bmatrix} P & B^\top \\ B & 0 \end{bmatrix}, \quad X_0 = \begin{bmatrix} Z_{\text{KKT}} \\ \Lambda \end{bmatrix}, \quad K_\delta = \begin{bmatrix} P + E_\delta & B^\top \\ B & -\delta W^{-1} \end{bmatrix}, \quad X_\delta = \begin{bmatrix} Z_\delta^{\text{plug}} \\ \eta_\delta^{\text{plug}} \end{bmatrix}.$$

Define the right-hand sides as

$$R^\star = \begin{bmatrix} G + (\partial_\theta B^\top) y^\star \\ g_\theta \end{bmatrix}, \quad R_\delta = \begin{bmatrix} G + (\partial_\theta B^\top) y^\star + F_\delta \\ g_\theta \end{bmatrix}.$$

Then (19) is $K_0 X_0 = -R^\star$, and (20) is $K_\delta X_\delta = -R_\delta$. Since

$$K_\delta - K_0 = \begin{bmatrix} E_\delta & 0 \\ 0 & -\delta W^{-1} \end{bmatrix},$$

we have $\|K_\delta - K_0\|_2 \le \|E_\delta\|_2 + \delta\|W^{-1}\|_2$. Since $K_0$ is nonsingular under LICQ and $P \succ 0$, let $M = \|K_0^{-1}\|_2$. Choose $\delta_0 > 0$ such that for all $0 < \delta \le \delta_0$, $M\big(\|E_\delta\|_2 + \delta\|W^{-1}\|_2\big) \le \frac{1}{2}$. Then the Neumann-series bound yields

$$\|K_\delta^{-1}\|_2 \le \frac{M}{1 - M\|K_\delta - K_0\|_2} \le 2M. \tag{21}$$

Now decompose $X_\delta - X_0 = (K_\delta^{-1} - K_0^{-1})(-R^\star) + K_\delta^{-1}\big(-(R_\delta - R^\star)\big)$. For the first term, using $K_\delta^{-1} - K_0^{-1} = K_0^{-1}(K_0 - K_\delta)K_\delta^{-1}$, we obtain

$$\|(K_\delta^{-1} - K_0^{-1})(-R^\star)\|_2 \le \|K_0^{-1}\|_2\|K_\delta - K_0\|_2\|K_\delta^{-1}\|_2\|R^\star\|_2 \le (2M^2\|R^\star\|_2)\big(\delta\|W^{-1}\|_2 + \|E_\delta\|_2\big).$$

For the second term, note that

$$R_\delta - R^\star = \begin{bmatrix} F_\delta \\ 0 \end{bmatrix},$$

hence by (21),

$$\|K_\delta^{-1}\big(-(R_\delta - R^\star)\big)\|_2 \le \|K_\delta^{-1}\|_2\|F_\delta\|_2 \le 2M\|F_\delta\|_2.$$

Combining the two bounds and using $\|E_\delta\|_2 \le \frac{c_E}{\delta}e^{-\gamma/\delta}$ and $\|F_\delta\|_2 \le \frac{c_F}{\delta}e^{-\gamma/\delta}$, we obtain

$$\|X_\delta - X_0\|_2 \le K_1\delta + \frac{K_2}{\delta}e^{-\gamma/\delta}, \tag{22}$$

for constants $K_1, K_2 > 0$ independent of $\delta$. Taking the primal block yields

$$\|Z_\delta^{\mathrm{plug}} - Z_{\mathrm{KKT}}\|_2 \le K_1\delta + \frac{K_2}{\delta}e^{-\gamma/\delta}. \tag{23}$$

Both terms on the right-hand side vanish as $\delta \to 0$, which implies that the plug-in sensitivity $Z_\delta^{\mathrm{plug}}$ satisfies $\|Z_\delta^{\mathrm{plug}} - Z_{\mathrm{KKT}}\|_2 \to 0$ as $\delta \to 0$. Furthermore, the auxiliary variable $\eta_\delta^{\mathrm{plug}} = \frac{1}{\delta}W(BZ_\delta^{\mathrm{plug}} + g_\theta)$ provides a consistent estimate of the KKT dual sensitivity $\Lambda = \partial_\theta y^\star$. The total block convergence $\|X_\delta - X_0\|_2 \to 0$ implies that $\|\eta_\delta^{\mathrm{plug}} - \Lambda\|_2 \to 0$ as $\delta \to 0$.

$\square$

# E. Experimental Details

All experiments were conducted on a machine with an Intel® Core™ Ultra 9 285H CPU and an NVIDIA GeForce RTX 5070 Laptop GPU (8 GB), using PyTorch 2.2 with CUDA 12.4.

## E.1. Projection Experiments

For each problem size, we summarize the mean runtime statistics over multiple random instances. In both dXPP and dQP, the Gurobi solver is run with an absolute residual tolerance of $\epsilon_{\mathrm{abs}} = 10^{-6}$, and the active set is identified using the threshold $\epsilon_{\mathcal{A}} = 10^{-5}$.

### E.1.1. PROJECTION ONTO THE PROBABILITY SIMPLEX

For each dimension $n$, the input $x \in \mathbb{R}^n$ is sampled with i.i.d. entries $x_i \sim \mathcal{N}(0, 1)$. We evaluate a dataset of 325 instances with

$$n \in \{20, 100, 450, 1000, 4600, 10000, 100000, 1000000\}.$$

To balance coverage at large scale, we generate 50 independent instances for each $n \le 4600$, and 25 instances for each $n > 4600$.

E.1.2. PROJECTION ONTO CHAINS

We fix the number of points at $m = 100$ and sample the inputs as $x_j \in \mathbb{R}^d$ with $x_j \sim \mathcal{N}(0, 100I_d)$. Letting $n = md$ denote the total number of primal variables, we vary $d$ to obtain

$$n \in \{200, 500, 1000, 2000, 4000, 10000, 100000, 1000000\}.$$

This yields 325 instances in total: we generate 50 independent problems for each $n \leq 4000$, and 25 problems for each $n > 4000$.

### E.2. End-to-End Multi-Period Portfolio Optimization

This section provides additional implementation details for the bilevel portfolio experiment in (14), including the rolling-window training procedure and the standard QP formulation of the inner problem.

The dataset consists of daily ETF prices over the period 2011–2024. We use daily returns of 7 tradable ETFs and rebalance the portfolio daily. For each decision date $t$, we form a feature window $X_t$ consisting of the past 120 days of ETF returns and predict a horizon-$H$ sequence of returns via a linear predictor. Following a standard rolling retraining pattern, the predictor is retrained periodically every 20 trading days using a look-back window of 120 supervised samples. Covariances are estimated in a rolling manner from 20-day historical returns prior to $t$ and are stabilized by adding a small diagonal ridge term to ensure positive definiteness. Each training epoch corresponds to one full pass through the chronological sequence of trading days in the dataset. Specifically, within each epoch, the following sequence of operations is performed for every decision date $t$: (i) the feature window $X_t$ is fed into the linear predictor to generate horizon-$H$ return forecasts $\hat{r}(\theta)$; (ii) the inner multi-period portfolio optimization problem (14) is solved to obtain the optimal weights $w^\star(\theta)$; (iii) the decision loss is computed by comparing the optimal weights against realized returns; (iv) gradients are backpropagated through the differentiable QP layer to update the model parameters $\theta$. The runtime results presented in Table 5 are calculated as the average over 100 training epochs.

Fix a decision date $t$ and horizon $H$. To express the inner portfolio optimization in (14) as a standard quadratic program, we introduce auxiliary variables $u_{t+k} \in \mathbb{R}^N$ to linearize the $\ell_1$ turnover constraints and define the augmented decision vector $\tilde{x} \in \mathbb{R}^{2NH}$. The resulting standard QP form is:

$$
\begin{aligned}
\min_{\tilde{x} \in \mathbb{R}^{2NH}} \quad & \frac{1}{2}\tilde{x}^\top P \tilde{x} + q(\theta)^\top \tilde{x} \\
\text{s.t.} \quad & A\tilde{x} = b, \quad G\tilde{x} \leq h \\
\text{where} \quad & \tilde{x} = \begin{bmatrix} w \\ u \end{bmatrix}, \quad P = \begin{bmatrix} \lambda\hat{\Sigma} & 0 \\ 0 & 0 \end{bmatrix}, \quad q(\theta) = \begin{bmatrix} -\hat{r}(\theta) \\ 0 \end{bmatrix} \\
& w = [w_{t+1}^\top, \dots, w_{t+H}^\top]^\top, \quad u = [u_{t+1}^\top, \dots, u_{t+H}^\top]^\top \\
& \hat{\Sigma} = \text{blkdiag}(\hat{\Sigma}_{t+1}, \dots, \hat{\Sigma}_{t+H}), \quad \hat{r}(\theta) = [\hat{r}_{t+1}(\theta)^\top, \dots, \hat{r}_{t+H}(\theta)^\top]^\top \\
\text{and constraints} \quad & \forall k \in \{1, \dots, H\} : \\
& \mathbf{1}^\top w_{t+k} = 1, \quad w_{t+k} \geq 0, \quad u_{t+k} \geq 0 \\
& -u_{t+k} \leq w_{t+k} - w_{t+k-1} \leq u_{t+k} \\
& \mathbf{1}^\top u_{t+k} \leq \tau
\end{aligned}
\tag{24}
$$

where $w_t$ is treated as the known pre-trade portfolio constant for the $k = 1$ turnover constraint. As formulated in (24), the complexity of the inner QP scales linearly with the investment horizon $H$. Each planning stage $k \in \{1, \dots, H\}$ introduces a weight vector $w_{t+k} \in \mathbb{R}^N$ and an auxiliary vector $u_{t+k} \in \mathbb{R}^N$, resulting in an augmented primal dimension of $\tilde{n} = 2NH$. Similarly, the number of constraints in $A$ and $G$ grows linearly with $H$: there are $H$ equality constraints for simplex normalization, and $O(NH)$ linear inequalities arising from the turnover bounds, the turnover budget $\mathbf{1}^\top u_{t+k} \leq \tau$, and the non-negativity constraints. This linear scaling makes the horizon $H$ the primary parameter for controlling the problem size in our dense-mode scalability study.

### E.3. Additional Experiments: Sudoku

This section reports additional experiments on the Sudoku benchmark. We follow the experimental setting in Amos & Kolter (2017, Sec. 4), where solving Sudoku is cast as a differentiable optimization layer. Concretely, each instance is a size-$n$

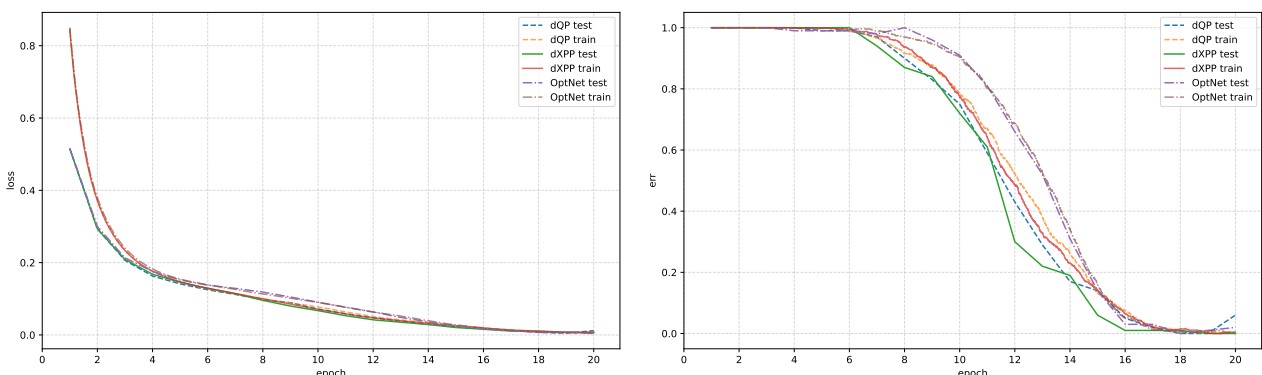

*Figure 4.* Sudoku experiments: training loss (left) and error rate (right) for dXPP, dQP, and OptNet.

*Table 6.* Average per-epoch runtime comparison for Sudoku experiments (ms).

| Board size $n$ | dXPP | dQP | OptNet |
|---|---|---|---|
| 2 | **12.15** | 12.81 | 12.16 |
| 3 | **196.21** | 261.86 | 217.04 |
| 4 | **2591.54** | 3109.08 | 7877.82 |

Sudoku puzzle with $n \in \{2, 3, 4\}$, corresponding to $4 \times 4$, $9 \times 9$, and $16 \times 16$ boards. At each training step, the model takes a puzzle as input and predicts per-cell scores over candidate digits. An optimization layer enforces Sudoku constraints and outputs a structured solution, and the loss is computed against the ground-truth solution. We compare dXPP with dQP and OptNet in both runtime and training dynamics.

**Training dynamics.** Figure 4 displays the evolution of both the training loss and the error rate over the course of training. The error rate is measured as the fraction of puzzles for which the predicted solution does not exactly match the ground truth. As shown, dXPP, dQP, and OptNet exhibit remarkably similar convergence behaviors: the loss functions decay at comparable rates, and the error trajectories follow nearly identical downward trends. This strong alignment confirms that the gradients produced by dXPP are as effective for learning structured constraints as those from established KKT-based differentiation methods, ensuring stable and reliable end-to-end training.

**Runtime efficiency.** Table 6 summarizes the average runtime per epoch. dXPP consistently outperforms the baselines across all board sizes. Notably, as the problem scale increases (e.g., for $n = 4$), the computational advantage of dXPP becomes increasingly significant, further validating its scalability in structured prediction tasks.

