# OpenReview forum: "A Penalty Approach For Differentiation Through Black-box Quadratic Programming Solvers"
_ICML.cc/2026/Conference — ICML 2026 regular_

### Official Review · Reviewer_aSar · 2026-03-12

**Soundness:** 4
**Presentation:** 3
**Significance:** 3
**Originality:** 3
**Overall Recommendation:** 5
**Confidence:** 4

**Summary:**

This paper investigates an important aspect of differentiable optimization layers: how to differentiate through black-box QP solvers without relying on KKT-based backward passes. The paper assesses the concept through a penalty-based reformulation, where the backward pass reduces to solving a primal-space SPD linear system rather than a larger indefinite KKT system. The overall idea is interesting, practically relevant, and well aligned with current interest in scalable differentiable optimization. The paper also provides both theory and experiments, including gradient-accuracy checks, scalability benchmarks, and an end-to-end portfolio optimization application.

**Compliance With Llm Reviewing Policy:**

Affirmed.

**Key Questions For Authors:**

1) Do the theoretical results or algorithmic guarantees continue to hold when P(\theta) is positive semidefinite rather than strictly positive definite? Some baseline methods, such as OptNet, require only positive semidefiniteness.

2) Could the authors clarify the role of the penalty strength in Algorithm 1? It would be helpful to explain how this parameter influences the optimization process, and whether the method is sensitive to its choice in practice or in theory.

**Limitations:**

yes

**Strengths And Weaknesses:**

Strengths:
1) The paper addresses an important problem in differentiable quadratic programming, namely the scalability and numerical fragility of KKT-based differentiation. The proposed dXPP framework is easy to understand at a high level and has a clear systems-level motivation. A key strength is the penalty-based backward mechanism, which replaces KKT differentiation with a primal-dimensional SPD linear system. This is a compelling design choice because it is naturally more compatible with sparsity and black-box solvers.

2) The paper does not present the method as purely heuristic; it also proves that the sensitivity obtained from the smoothed penalty formulation converges to the exact KKT sensitivity as the smoothing parameter tends to zero. This gives the method a strong theoretical foundation.

3) The experiments suggest strong backward-pass scalability, with dXPP outperforming SOTA dQP at large problem sizes and remaining feasible in regimes where some prior methods do not complete. The reported large-scale results are a meaningful empirical strength.

Weaknesses:

1) Sensitivity to hyperparameters is not sufficiently investigated. The performance of dXPP may be sensitive, both in accuracy and runtime, to the choice of the smoothing parameter \delta and the penalty strength \zeta. Since these parameters appear central to the practical behavior of the method, the paper should provide a more systematic sensitivity analysis and report how robust the approach is across different settings.

2) Accuracy metrics for calculating gradients are not sufficiently specified. My main concern is that the accuracies of the proposed method dXPP and the baseline methods are not fully specified or consistently reported in a way that makes comparison easy to interpret (Tables 2-3). Furthermore, while the paper evaluates relative gradient discrepancy and runtime, it would be helpful to more clearly report downstream task accuracy/solution quality for dXPP versus all baselines, especially in the application-level end-to-end experiments (Table 4). This omission makes it somewhat harder to judge the full tradeoff between computational efficiency and predictive/optimization performance

---

> ### Author Rebuttal · Authors · 2026-03-31
>
> > **W1. Sensitivity to hyperparameters is not sufficiently investigated.**
>
> Thank you for this comment. For the sensitivity w.r.t. $\zeta$, please see our reply to **Q2**. For $\delta$, please refer to `W3 & Q1 response to Reviewer a7UL`.
>
> > **W2. Gradient accuracies are not fully specified.(Table 2-3); downstream task solution quality is desirable (Table 4).**
>
> We agree that the presentation of accuracy across methods was not sufficiently uniform in the original version. In the revised manuscript, we have standardized the reporting format to make the comparisons easier to interpret. In particular, we now state explicitly that the gradient-accuracy metric is the relative error
> $\epsilon_{\rm rel}=\frac{\lVert g_{\rm dXPP}-g_{\rm ref}\rVert_2}{\lVert g_{\rm ref}\rVert_2}$
> , and we use this definition consistently when discussing gradient accuracy results. We plan to revise the text and table presentation to make clear which quantities correspond to accuracy and which correspond to runtime, so that the comparison between dXPP and the baselines is more direct.
>
> The issue of downstream task solution quality is also raised by Reviewer 6zzS regarding end-to-end portfolio quality, so we address it with additional experiments (see `W4 & Q2 response to Reviewer 6zzS`).
>
> > **Q1. Do the theory applies when P is PSD rather than strictly PD? Some baseline methods, (e.g. OptNet), require only PSD.**
>
> This is a great question. Our theorem uses $P \succ 0$ as a simple sufficient condition, but it can be relaxed to the second-order sufficient condition (SOSC): $v^\top P v > 0$ for all nonzero $v$ with $Bv = 0$, i.e., $\ker(P) \cap \ker(B) = \{0\}$. Under LICQ, strict complementarity, and SOSC, our proof extends directly. Without SOSC, the primal solution is non-unique and sensitivities are ill-defined for any method. We will state this relaxed condition in the revision.
>
> Regarding baselines: although OptNet states $P \succeq 0$ in its formulation, its Theorem 1 assumes $P \succ 0$ for well-definedness of $\partial z^\star(\theta)$. dQP likewise requires positive definiteness explicitly. SOSC is a necessary condition at the problem level, and all existing methods require it in some form.
>
> > **Q2. "Clarify the role of the penalty strength ζ, its sensitivity in practice or in theory.**
>
> Thank you for the question. Once $\zeta \ge 1$ satisfies the exact-penalty threshold (Proposition 3.1), its main effect is to rescale the regularization jointly with $\delta$: in the equivalent block system, the perturbation is $-\delta W^{-1}$, which scales as $O(\delta/\zeta)$. Thus $\zeta$ acts as a moderate safety factor rather than a sensitive hyperparameter. In practice we fix $\zeta=10$ throughout and do not tune it; if any tuning is needed, $\delta$ is the more important parameter.
>
> Our sensitivity study with $\zeta \in \{1,6,8,10,100\}$ confirms this: extreme values ($\zeta=1$ or $100$) slightly degrade gradient accuracy, while moderate values ($\zeta=6,8,10$) perform consistently well. Backward runtime is essentially unaffected (e.g., 226–239 ms at size $10^6$).
>
> **Effect of $\zeta$ on gradient discrepancy**
>
> | QP size | $\zeta=1$ | $\zeta=6$ | $\zeta=8$ | $\zeta=10$ | $\zeta=100$ |
> |---|---:|---:|---:|---:|---:|
> | $10 * 5$ | $2.72 * 10^{-6}\pm1.67 * 10^{-6}$ | $3.53 * 10^{-7}\pm2.41 * 10^{-7}$ | $2.37 * 10^{-7}\pm1.55 * 10^{-7}$ | $\mathbf{1.91 * 10^{-7}\pm1.40 * 10^{-7}}$ | $2.32 * 10^{-7}\pm2.11 * 10^{-7}$ |
> | $100 * 20$ | $1.06 * 10^{-5}\pm2.52 * 10^{-5}$ | $\mathbf{1.32 * 10^{-7}\pm5.21 * 10^{-8}}$ | $1.84 * 10^{-7}\pm4.93 * 10^{-8}$ | $2.64 * 10^{-7}\pm9.50 * 10^{-8}$ | $2.52 * 10^{-6}\pm9.67 * 10^{-7}$ |
> | $1000 * 200$ | $1.79 * 10^{-4}\pm2.32 * 10^{-4}$ | $\mathbf{1.03 * 10^{-5}\pm9.66 * 10^{-7}}$ | $1.37 * 10^{-5}\pm1.24 * 10^{-6}$ | $1.74 * 10^{-5}\pm2.25 * 10^{-6}$ | $1.62 * 10^{-4}\pm2.20 * 10^{-5}$ |
> | $5000 * 2000$ | $6.12 * 10^{-3}\pm3.92 * 10^{-3}$ | $\mathbf{4.24 * 10^{-4}\pm1.03 * 10^{-5}}$ | $4.67 * 10^{-4}\pm1.88 * 10^{-5}$ | $5.14 * 10^{-4}\pm2.39 * 10^{-5}$ | $5.33 * 10^{-3}\pm4.95 * 10^{-4}$ |
>
> **Effect of $\zeta$ on backward runtime (ms)**
>
> | Problem size | $\zeta=1$ | $\zeta=6$ | $\zeta=8$ | $\zeta=10$ | $\zeta=100$ |
> |---|---:|---:|---:|---:|---:|
> | $100$ | $0.77$ | $0.84$ | $0.82$ | $0.74$ | $0.75$ |
> | $4600$ | $1.11$ | $1.08$ | $1.09$ | $1.10$ | $1.13$ |
> | $100000$ | $8.70$ | $9.06$ | $8.35$ | $8.69$ | $9.23$ |
> | $1000000$ | $235.12$ | $237.47$ | $226.21$ | $239.03$ | $237.17$ |

---

> > ### Author Rebuttal · Reviewer_aSar · 2026-04-03
> >
> > Thank you for the response. The authors have addressed my concerns. Overall, I will keep my score.

---

> > > ### Author Response · Authors · 2026-04-05
> > >
> > > Thank you for your thoughtful review and supportive assessment of our work. We especially appreciate your recognition of both the motivation and the technical contribution of the paper.

---

### Official Review · Reviewer_6zzS · 2026-03-12

**Soundness:** 3
**Presentation:** 4
**Significance:** 4
**Originality:** 3
**Overall Recommendation:** 5
**Confidence:** 4

**Summary:**

This paper proposes dXPP, a way to differentiate through convex QPs without differentiating through the KKT system. The forward solve is handled by any black-box QP solver. For the backward pass, dXPP reformulates the QP as a smoothed exact penalty problem and differentiates through that, which only requires solving an $n \times n$ SPD system in the primal variables (instead of a larger indefinite KKT system). The method does not actually solve the penalty problem: it evaluates the penalty Hessian at the original QP solution $z^\star$ ("plug-in" sensitivity) and shows this converges to the true KKT sensitivity as the smoothing $\delta \to 0$ (Theorem 3.4), assuming LICQ and strict complementarity. Experiments on random QPs, large sparse projections (up to $10^6$ variables), and a portfolio optimization task show that dXPP matches gradient accuracy of KKT-based methods with speedups up to $9\times$ in the backward pass.

**Compliance With Llm Reviewing Policy:**

Affirmed.

**Final Justification:**

The paper has a clean idea and solid theory; the main gaps were experimental, and the rebuttal fills them convincingly.

**Key Questions For Authors:**

1. **Parameter sensitivity:** What happens when $\delta$ is larger (e.g., $10^{-3}$, $10^{-2}$)? Does accuracy degrade gracefully? When $\delta$ is very small, does the conditioning of $H$ cause problems? Tradeoff curves would help a lot here.
2. **End-to-end quality:** In the portfolio experiment, do dXPP, dQP, and OptNet produce models of comparable quality (realized returns, decision loss)? If yes, the speedups are compelling. If not, the runtime comparison is incomplete.
3. **Gradient error scaling:** The error in Table 1 grows monotonically with problem size. Is this from the penalty approximation or from solver tolerance? Would tighter solver tolerances help at larger scales?

**Limitations:**

Partially. The conclusion mentions the restriction to convex QPs and points to broader convex programs as future work. It would be useful to also discuss: (1) the reliance on LICQ and strict complementarity for the convergence guarantee, and what can be said when these fail, (2) the monotonically growing gradient error with problem size (Table 1), and (3) the gap between runtime benchmarks and end-to-end learning quality in the portfolio experiment. Societal impact is not a concern for this work.

**Strengths And Weaknesses:**

**strengths:**

1. The penalty reformulation with softplus smoothing is a nice idea. Reducing the backward pass from an indefinite $(n+p+|\mathcal{A}|)$ KKT system to an $n \times n$ SPD system is a real practical win, and the benchmarks across increasing dimensions show this clearly.
2. The plug-in trick (Section 3.3) is clever: instead of solving the smoothed penalty problem, just evaluate the penalty Hessian at the original QP solution $z^\star$. Theorem 3.4 gives a clean justification.
3. Good experimental coverage. Scalability to $10^6$ variables (Tables 2-3) and the $343\times$ speedup over dQP in the portfolio experiment (Table 4, $H=200$) are convincing.
4. Well written. The flow from KKT differentiation (Section 3.1) to penalty reformulation (3.2) to plug-in analysis (3.3) to computational discussion (3.4) is easy to follow.




**weaknesses:**

1. **No sensitivity analysis for $\delta$ and $\zeta$.** A single setting ($\delta = 10^{-6}$, $\zeta = 10$) is used everywhere. The bound (eq. 28) shows that smaller $\delta$ improves accuracy but makes $H = P + \frac{1}{\delta} B^\top W B$ more ill-conditioned. The paper should show tradeoff curves for gradient accuracy and runtime vs. $\delta$ and $\zeta$.
2. **Robustness claims go beyond the theory.** Theorem 3.4 needs LICQ and strict complementarity, but Section 3.4 claims dXPP works even when these fail. The argument is that $H$ stays SPD as long as $P \succ 0$, which is true, but there is no reason to believe the resulting gradient is useful in degenerate cases. The portfolio experiment is the natural test for this (many active constraints, frequent degeneracy), but it only reports runtime, not gradient quality or model performance.
3. **Gradient error grows with problem size.** Table 1 shows relative error going from $\sim 10^{-7}$ (small QPs) to $\sim 10^{-4}$ ($5000 \times 2000$). The paper says this "remains below $10^{-3}$," but the trend is monotone and unexplained. Is this from the penalty approximation itself, or from solver tolerance? This matters for the scalability story.
4. **No end-to-end quality comparison.** The portfolio experiment (Section 4.3) only reports training time per epoch. It does not compare the quality of the learned models (decision loss, realized returns) across methods. Faster gradients are only useful if they lead to comparable models.


**minor comments:**
- BPQP (Pan et al., 2024) is cited in Section 4.1 for the random QP benchmark but missing from Tables 2-4. It would be good to include it.
- The active-set pruning heuristic (dropping $E_\delta$ and $F_\delta$, end of Section 3.4) is quite different from the theory presented. It would help to show how much this affects gradient accuracy.
- The literature on differentiating through general conic programs (Agrawal et al., 2019b; Busseti et al., 2019) could be discussed in terms of whether the penalty approach extends beyond QPs to SOCPs or SDPs.

---

> ### Author Rebuttal · Authors · 2026-03-31
>
> > **W1, Q1. Sensitivity to $\delta$ and $\zeta$.**
>
> Thank you. For $\delta$, please see `W3/Q1 response to Reviewer a7UL`; for $\zeta$, `Q2 response to Reviewer aSar`.
>
> > **Q1. Large/small $\delta$?**
>
> We additionally tested larger and smaller $\delta$. As suggested by our scale-aware rule $\delta \propto \rho\|KKT\|$, the trend depends on scale. For a small QP ($50\times10$), accuracy improves as $\delta$ decreases, so large smoothing mainly introduces approximation bias. For a large QP ($5000\times2000$), making $\delta$ too small eventually hurts accuracy, showing a trade-off between approximation error and the conditioning of $H$.
>
> | size | $10^{-2}$ | $10^{-3}$ | $10^{-4}$ | $10^{-5}$ | $10^{-6}$ |
> |---|---:|---:|---:|---:|---:|
> | $50*10$ | $3.00*10^{-4}$ | $4.24*10^{-5}$ | $2.72*10^{-6}$ | $3.63*10^{-7}$ | $8.55*10^{-8}$ |
> | $5000*2000$ | $\mathbf{2.62*10^{-7}}$ | $5.11*10^{-7}$ | $5.18*10^{-6}$ | $5.08*10^{-5}$ | $5.14*10^{-4}$ |
>
> > **W2. Robustness under degeneracy.**
>
> We agree the exact KKT sensitivity is not uniquely defined under degeneracy. Our claim is instead that the dXPP gradient remains well-motivated and numerically stable. As shown in Appendix C, Eq. (13) is equivalent to Eq. (25), where the zero KKT block is replaced by $-\delta W^{-1}$. Eliminating the dual variable gives $(P + \frac{1}{\delta} B^\top W B) Z_\delta = \text{RHS}$ which has the form of a Tikhonov-regularized normal equation: $\delta^{-1}B^\top W B$ enforces constraint satisfaction, while $P$ selects a stable solution when $B$ is rank-deficient (LICQ fails). This explains why dXPP remains well-defined, whereas KKT-based methods face singular systems and often require damping or least-squares fixes (as in dQP). A formal characterization under degeneracy is future work.
>
> Empirically, in response to **W4&Q2**, we added an end-to-end portfolio comparison and found that dXPP yield better returns than dQP, Sharpe ratios, and drawdowns, confirming that dXPP gradients remain effective for learning in this degeneracy-prone setting.
>
> > **W3, Q3. Why does gradient error increase with problem size? Does solver tolerance matter?**
>
> To separate solver accuracy from smoothing error, we varied the solver tolerance while fixing all other settings. The gradient discrepancy changed little, so the monotone increase in Table 1 is not mainly due to solver tolerance; the dominant source is the smoothing approximation in the backward pass.
>
> | size | $10^{-5}$ | $10^{-6}$ | $10^{-7}$ | $10^{-8}$ | $10^{-9}$ |
> |---|---:|---:|---:|---:|---:|
> | $10*5$ | $2.54*10^{-7}$ | $1.18*10^{-7}$ | $1.49*10^{-7}$ | $1.91*10^{-7}$ | $2.11*10^{-7}$ |
> | $100*20$ | $2.46*10^{-7}$ | $2.41*10^{-7}$ | $1.92*10^{-7}$ | $2.64*10^{-7}$ | $3.30*10^{-7}$ |
> | $1000*200$ | $1.68*10^{-5}$ | $1.79*10^{-5}$ | $1.79*10^{-5}$ | $1.74*10^{-5}$ | $1.84*10^{-5}$ |
> | $5000*2000$ | $5.16*10^{-4}$ | $5.31*10^{-4}$ | $5.21*10^{-4}$ | $5.09*10^{-4}$ | $5.19*10^{-4}$ |
>
> > **W4, Q2. End-to-end model quality.**
>
> We added an end-to-end portfolio comparison on the last 18 months of data (2023/06--2024/12). dXPP and dQP achieve comparable learning quality, while dXPP is far faster in the backward pass. dXPP also slightly improves return, Sharpe ratio, and max drawdown over dQP, and clearly outperforms PTO and OptNet. Because of rebuttal-time limits, we ran this as a focused validation and will extend it in the revision.
>
> | Method | Return | Vol | Sharpe | MDD | Backward |
> |---|---:|---:|---:|---:|---:|
> | dXPP | **0.0838** | **0.1166** | **0.7181** | **-0.0545** | **238 ms** |
> | dQP | 0.0814 | 0.1161 | 0.7014 | -0.0560 | 29731 ms |
> | OptNet | 0.0532 | 0.1178 | 0.4517 | -0.0722 | 583 ms |
> | PTO | 0.0738 | 0.1184 | 0.6236 | -0.0720 | -- |
>
> > **C1. BPQP missing from Tables 2--4.**
>
> We were unable to include BPQP because the code released through Qlib is currently unavailable from the project page. We will include it once the implementation becomes accessible.
>
> > **C2. Effect of active-set pruning.**
>
> We added an ablation comparing the pruned version (dropping $E_\delta,F_\delta$) with the full version. Pruning does not hurt gradient accuracy and slightly improves it, consistent with these terms vanishing asymptotically but introducing small numerical noise at finite $\delta$.
>
> | size | pruning | full |
> |---|---:|---:|
> | $10*5$ | $2.84*10^{-7}$ | $3.12*10^{-7}$ |
> | $100*20$ | $2.09*10^{-7}$ | $2.30*10^{-7}$ |
> | $1000*200$ | $1.25*10^{-5}$ | $1.38*10^{-5}$ |
> | $5000*2000$ | $3.51*10^{-4}$ | $3.86*10^{-4}$ |
>
> > **C3. Extension to conic programs.**
>
> Our framework is not limited to QPs. For general conic constraints, one may penalize distance to the cone and apply the same smoothing-and-differentiation strategy. The main requirements remain uniqueness and regularity of the solution mapping. We are actively extending the method to SOCPs and SDPs and plan to report these results in follow-up work.

---

> > ### Author Rebuttal · Reviewer_6zzS · 2026-04-02
> >
> > The authors have addressed all my concerns. The sensitivity studies for $\delta$ and $\xi$ (W1/Q1) show graceful tradeoffs and confirm the default setting is reasonable. The gradient error scaling (W3/Q3) is now explained as coming from the smoothing approximation, not solver tolerance. Most importantly, the end-to-end portfolio comparison (W4/Q2) shows dXPP matches or exceeds dQP in model quality while being ~125x faster in the backward pass, which completes the practical case for the method. I raise my score.

---

> > > ### Author Response · Authors · 2026-04-05
> > >
> > > We are deeply grateful for your insightful review, as well as for revisiting the paper after reading our rebuttal.  Thank you again for your thoughtful engagement with our work and for your willingness to raise the score.

---

### Official Review · Reviewer_Z1VF · 2026-03-13

**Soundness:** 3
**Presentation:** 2
**Significance:** 3
**Originality:** 2
**Overall Recommendation:** 4
**Confidence:** 3

**Summary:**

The authors consider constrained quadratic optimization problems as differentiable layers in neural networks. Prior approaches for backpropagating through such layers require solving a linear system whose size equals the combined dimension of the primal and dual variables. The authors propose a penalty-based formulation that incorporates both equality and inequality constraints into the objective, which reduces the dimension of the system involved in the backward pass to that of the primal variables only. They then apply implicit differentiation to this penalized objective. Although the resulting derivative of the solution is only an approximation of the true gradient, the authors show that the estimated derivative converges to the true derivative as the penalty parameter tends to zero. Experiments demonstrate improved computational performance compared to previous approaches.

**Compliance With Llm Reviewing Policy:**

Affirmed.

**Final Justification:**

The authors have addressed my concerns and therefore I am updating my score.

**Key Questions For Authors:**

I have following questions:
* How does the conditioning of the matrix $B$ in (4), affects the stability and computational cost of the backward pass for a good choice of $\delta>0$ for hypergradient accuracy as compared to the previous methods?
* What would be the effect of using warm-starting strategies [1, 2] when computing the backward pass for the methods compared in Tables 2, 3 and 4? Would the proposed method still outperform the previous methods by a significant margin?

**References**

[1] Ji, K., Yang, J. and Liang, Y., 2021, July. Bilevel optimization: Convergence analysis and enhanced design.

[2] Arbel, M. and Mairal, J., 2021. Amortized implicit differentiation for stochastic bilevel optimization.

**Limitations:**

Not applicable.

**Strengths And Weaknesses:**

**Strengths**
* The proposed approach reduces the dimension of the linear system required in the backward pass from the combined dimension of primal and dual variables to only the dimension of the primal variables.
* The work is well motivated and the proposed idea is simple but original. The penalty approach is used only to reduce the size of the linear system required for evaluating the backward pass, while the forward pass can still be computed using any efficient solver.
* The experimental section compares the proposed method with several existing approaches, which helps position the work relative to prior literature.

**Weaknesses**
* The accuracy of the proposed hypergradient depends on the penalty parameter $\delta$ being sufficiently small. However, for poorly conditioned matrices $A$ and $C_{\mathcal A}$, the system matrix may become ill-conditioned due to the presence of terms proportional to $1 / \delta$. This could potentially slow down the backward pass. The paper does not investigate such cases experimentally, nor do they provide any comment.
* Some parts of the presentation are unclear. In particular, Section 3.3, which contains the main contribution of the paper, introduces several new quantities and notations without sufficient motivation, which may cause confusion for readers.
* In Proposition 3.3, the matrices $E_\delta$ and $F_\delta$ are not clearly defined. The statement that the associated terms vanish as $\delta \to 0$ is therefore difficult to interpret. Although some intuition is provided later in Section 3.4 under the heading *Support for active-set pruning*, this explanation appears too late.
* There are also minor presentation issues. For example, the matrix $H$ is used in Line 243 before being formally defined (Line 260). Additionally, the statement in Line 217 that “these multipliers are used only to set the penalty magnitudes” appears inaccurate since the multipliers also appear in equation (13). Finally, the authors use three different notations for derivative, i.e., $\partial$ (Line 145), $(\cdot)^\prime (Line 174), and $\nabla$ (Line 214) and $\partial$ is used to denote the derivative and the subdifferential without any clarification.

---

> ### Author Rebuttal · Authors · 2026-03-30
>
> We thank the reviewer for the thorough feedback. We address each point below with new experiments and planned presentation improvements.
>
> > **W1. The system can be ill-conditioned due to 1/δ. How does the conditioning affect the stability and cost of the backward pass?**
>
> Thank you for the insightful comment. We agree that the conditioning deserves more explicit discussion. As you suggested, the practical question is how much it affects dXPP for a fixed $\delta>0$ chosen for good hypergradient accuracy.
>
> To quantify the practical impact, we fixed $\delta=10^{-4}$ and progressively made constraint rows nearly linearly dependent, increasing the KKT condition number. As expected, gradient discrepancy grows with conditioning, but only gradually: when the average KKT condition number increases from $10^4$ to $10^{10}$,  the average relative error rises from $5.38\times10^{-7}$ to $1.96\times10^{-4}$. The backward runtime stays essentially flat at about 130–166 ms throughout. Poor conditioning mainly affects accuracy, not runtime. We will add this experiment to the revision.
>
> | Avg. cond(KKT) | Avg. $\epsilon_{\mathrm{rel}}$ | Std. $\epsilon_{\mathrm{rel}}$ | dXPP bwd (ms) |
> | -------------: | -----------------------------: | -----------------------------: | ------------: |
> |  $1.76*10^{4}$ |                 $1.74*10^{-8}$ |                 $1.08*10^{-8}$ |        $131$ |
> |  $2.69*10^{4}$ |                 $3.11*10^{-8}$ |                 $8.52*10^{-9}$ |        $145$ |
> |  $9.69*10^{4}$ |                 $5.60*10^{-8}$ |                 $2.15*10^{-8}$ |        $166$ |
> |  $2.19*10^{6}$ |                 $3.50*10^{-7}$ |                 $3.58*10^{-7}$ |        $130$ |
> |  $8.73*10^{6}$ |                 $1.56*10^{-7}$ |                 $5.63*10^{-7}$ |        $143$ |
> |  $2.17*10^{8}$ |                 $2.74*10^{-6}$ |                 $6.00*10^{-6}$ |        $155$ |
> |  $8.27*10^{8}$ |                 $4.64*10^{-6}$ |                 $9.76*10^{-6}$ |        $163$ |
> |  $2.02*10^{10}$ |                 $1.33*10^{-5}$ |                 $5.82*10^{-5}$ |        $153$ |
> | $8.47*10^{10}$ |                 $1.35*10^{-5}$ |                 $7.45*10^{-5}$ |        $144$ |
>
> > **W2, W3. Section 3.3 introduces new quantities without sufficient motivation. E_δ and F_δ are not clearly defined.**
>
> We will restructure Section 3.3 by deriving $\nabla^2_{zz}\Phi_\delta$ and $\nabla^2_{z\theta}\Phi_\delta$ in the main text (rather than the appendix), which directly motivates the definitions of $E_\delta$ and $F_\delta$ as inactive-constraint contributions that decay as $\delta\to 0$. The derivation will be separated into clearer steps: first introducing the active-constraint objects $g, B, g_\theta, W$, then presenting the Hessian and mixed-derivative terms in separate propositions, and finally assembling the plug-in system and convergence statement.
>
> > **W4. Presentation issues.**
>
> We will fix all three issues: (1) swap the paragraphs so $H$ is defined before use; (2) correct Line 217 to clarify that multipliers also enter the plug-in term $(\partial_\theta B^\top)y^\star$ in Eq. (13); (3) use $\nabla$ for optimization-variable gradients, $\partial_\theta$ for parameter derivatives, and reserve primes for scalar smoothing functions $p_\delta', p_\delta''$.
>
> > **Q1. How does the conditioning of $B$ affect stability and cost?**
>
> See **W1**.
>
> > **Q2. What would be the effect of warm-starting strategies [1, 2] for Tables 2, 3, 4?**
>
> The reviewer proposes two interesting questions. First, warm-starting is mainly relevant to Table 4, not Tables 2–3 which benchmark one-shot backward runtime. Moreover, [1,2] target smooth unconstrained strongly-convex lower-level problems, whereas our layer involves constrained QPs with active-set changes and possible degeneracy. In decision-focused learning, distinct QPs are solved across different samples or decision dates, so one instance's solution does not serve as a good initializer for the next.
>
> We still tested warm-started CG for both dQP and dXPP on a rolling window of portfolio optimization. Warm-start helps both methods but does not remove dXPP's advantage: at $H$=200, dXPP+warm-CG (57.46 ms) remains ~6$\times$ faster than dQP+warm-CG (352.68 ms). This confirms that dXPP's speedup stems from the smaller SPD backward system, a structural advantage that warm-starting does not eliminate.
>
> | Horizon $H$ |   dQP-CG | dQP+warm |  dXPP-CG | dXPP+warm |
> | ----------- | -------: | -------: | -------: | --------: |
> | $10$        |  $17.38$ |   $3.68$ |   $5.56$ |    $1.97$ |
> | $20$        |  $19.75$ |   $8.03$ |   $6.93$ |    $2.47$ |
> | $50$        |  $51.82$ |  $34.01$ |  $10.95$ |    $6.15$ |
> | $100$       | $262.95$ |  $96.52$ |  $40.65$ |   $23.32$ |
> | $150$       | $533.58$ | $227.67$ |  $70.79$ |   $37.38$ |
> | $200$       | $886.93$ | $352.68$ | $104.17$ |   $57.46$ |
>
> ---
>
> If these additional results address your concerns, we would be grateful if you could consider updating your score accordingly.

---

> > ### Author Rebuttal · Reviewer_Z1VF · 2026-04-04
> >
> > I want to thank the authors for their thoughtful rebuttal and for conducting the additional experiments. I also appreciate the authors’ willingness to incorporate the suggested improvements to the presentation. These clarifications and additions address my main concerns, and I am therefore revising my score from Weak Reject to Weak Accept.

---

> > > ### Author Response · Authors · 2026-04-05
> > >
> > > We are very grateful that you found the main concerns adequately addressed and updated the score!  Your feedback was especially valuable in helping us strengthen both the technical presentation and the empirical support of the paper.

---

### Official Review · Reviewer_a7UL · 2026-03-13

**Soundness:** 3
**Presentation:** 2
**Significance:** 3
**Originality:** 2
**Overall Recommendation:** 4
**Confidence:** 3

**Summary:**

The paper studies differentiation through Quadtic Programs and proposes dXPP, a differentiation method that incorporates the constraints in the objective using a Lagrangian with smoothed penalty constraints. This reduces the costs of previous methods, such as differentiating through the KKT conditions and scales better to larger QPs, as shown by experiments.

**Compliance With Llm Reviewing Policy:**

Affirmed.

**Final Justification:**

See Rebuttal acknowledgment.

**Key Questions For Authors:**

1 - Can you provide an experiment that shows the effect of the choice of $\delta$?

2 - What happens without smoothing?

3 - What if we choose a different kind of smoothing?

4 - How fast does the sensitivity as a function of $\deltta$ converge to the true sensitivity?

**Limitations:**

Other smoothing strategies, no smoothing?

general problems.

**Strengths And Weaknesses:**

Soundness: The results and derivations all look correct.

Presentation: overall good. Some notations need to be clarified, such as not explicitly writing $\theta$ in the different matrices; also, quantities like $E_\delta$ and $F_\delta$ have not been explicitly defined.

Significance: I am not very familiar with settings where such problems are needed, but the experiments show that the computational gains from the method are very significant.

Originality: Besides the choice of the smoothing, applying the implicit function theorem to the Lagrangian seems to be a natural idea. The smoothing choice was not fully motivated, in my opinion (why use it? Why not something else?), and the experiments neglected to show the effect of the smoothing parameter $\delta$; this point also applies to the presentation.

---

> ### Author Rebuttal · Authors · 2026-03-31
>
> > **W1. (Presentation) Some notations need to be clarified, such as not explicitly writing θ in the different matrices; also, quantities like E_δ and F_δ have not been explicitly defined.**
>
> Thank you for the suggestion. In the revision, we will clarify that $P,A,C,q,b,d$ depend on $\theta$, and we omit the argument when there is no ambiguity. The definitions of $E_\delta$ and $F_\delta$ are currently given in Eqs. (15) and (17) of the appendix; we will move them to the main text.
>
> > **W2, Q3. The smoothing choice was not fully motivated (why use it? Can you use other smoothing strategies?)**
>
> The key requirement on the smoothing $p_\delta(t)$ is that it is **twice continuously differentiable** at the kink $t=0$, so that $\nabla^2_{zz}\Phi_\delta$ and $\nabla^2_{z\theta}\Phi_\delta$ are well defined and Eq. (10) is valid. This rules out options such as Huber, which is only $C^1$. Among smooth choices, softplus is natural: it is a standard soft approximation of $\max\{0,t\}$, has stable and efficient derivatives in autodiff frameworks, and is also supported by prior theory on softplus penalties for large-scale convex optimization (Li et al., 2023).
>
> > **W2, Q2. What happens without smoothing?**
>
> Without smoothing, the exact penalty is nonsmooth and yields a stationarity inclusion rather than a smooth equation, due to the subdifferentials of the $\ell_1$ norm and the hinge function $[\cdot]_+$. Implicit differentiation therefore requires selecting a particular subgradient. The natural KKT-consistent choice is
>
> $$
> 0 = \nabla f(z^\star) + \rho A^\top s^\star + \alpha C_{\mathcal{A}}^\top t_{\mathcal{A}}^\star,\quad
> Az^\star = b,\quad
> C_{\mathcal{A}} z^\star = d_{\mathcal{A}}.
> $$
>
> However, differentiating this system under a fixed active set simply recovers the reduced KKT system (Eq. 4). Thus, the nonsmooth route does not provide a new backward method; it reduces to standard KKT differentiation, with the same indefinite systems and sensitivity issues that we aim to avoid.
>
> > **W3, Q1. The experiments neglected to show the effect of the smoothing parameter δ. Can you provide an experiment that shows the effect of the choice of δ?**
>
> Thank you for pointing this out. We added an explicit sensitivity study on $\delta$. To compare instances of different scales, we choose $\delta$ by a scale-aware rule, $\delta \propto \rho \|KKT\|$, where $\|KKT\|$ denotes the norm of the KKT matrix appearing on the left-hand side of Eq. (4), rounded to the nearest power of ten.
>
> Across all tested sizes, the best accuracy is attained at an intermediate smoothing level: large $\delta$ introduces smoothing bias, while very small $\delta$ makes the backward system more numerically sensitive. By contrast, backward runtime is much less sensitive to $\delta$, so $\delta$ mainly affects accuracy rather than cost.
>
> **Effect of $\rho$ on gradient accuracy (choosing $\delta \propto \rho \|KKT\|$)**
>
> | QP size | KKT norm | $\rho=10^{-5}$ | $\rho=10^{-6}$ | $\rho=10^{-7}$ | $\rho=10^{-8}$ | $\rho=10^{-9}$ |
> |---|---:|---:|---:|---:|---:|---:|
> | $10 * 5$ | $4.92 * 10^{1}$ | $3.67 * 10^{-5}\pm3.38 * 10^{-5}$ | $1.48 * 10^{-6}\pm7.51 * 10^{-7}$ | $\mathbf{1.91 * 10^{-7}\pm1.40 * 10^{-7}}$ | $9.07 * 10^{-7}\pm6.54 * 10^{-7}$ | $1.11 * 10^{-5}\pm7.42 * 10^{-6}$ |
> | $100 * 20$ | $1.43 * 10^{3}$ | $1.07 * 10^{-6}\pm2.45 * 10^{-7}$ | $2.03 * 10^{-7}\pm6.00 * 10^{-8}$ | $\mathbf{1.03 * 10^{-7}\pm1.60 * 10^{-8}}$ | $2.61 * 10^{-7}\pm1.03 * 10^{-7}$ | $3.02 * 10^{-6}\pm1.94 * 10^{-6}$ |
> | $1000 * 200$ | $4.47 * 10^{4}$ | $8.83 * 10^{-7}\pm1.70 * 10^{-7}$ | $2.78 * 10^{-7}\pm5.10 * 10^{-8}$ | $\mathbf{1.83 * 10^{-7}\pm1.19 * 10^{-8}}$ | $1.58 * 10^{-6}\pm2.03 * 10^{-7}$ | $1.12 * 10^{-5}\pm1.37 * 10^{-5}$ |
> | $5000 * 2000$ | $5.00 * 10^{5}$ | $3.18 * 10^{-6}\pm2.40 * 10^{-7}$ | $7.25 * 10^{-7}\pm6.10 * 10^{-8}$ | $\mathbf{2.62 * 10^{-7}\pm4.56 * 10^{-9}}$ | $2.11 * 10^{-6}\pm1.52 * 10^{-7}$ | $1.63 * 10^{-5}\pm1.21 * 10^{-5}$ |
>
> **Effect of $\delta$ on backward runtime (ms)**
>
> | Problem size | $\delta=10^{-2}$ | $\delta=10^{-4}$ | $\delta=10^{-6}$ | $\delta=10^{-8}$ |
> |---|---:|---:|---:|---:|
> | $100$ | $0.72$ | $0.79$ | $0.82$ | $0.81$ |
> | $4600$ | $1.05$ | $1.11$ | $1.09$ | $1.22$ |
> | $100000$ | $8.51$ | $8.56$ | $8.35$ | $9.00$ |
> | $1000000$ | $236.40$ | $235.76$ | $226.21$ | $242.21$ |
>
> > **Q4. How fast does the sensitivity as a function of δ converge to the true sensitivity?**
>
> We added a sensitivity study on a fixed $50*10$ instance and measured the gradient error for different $\delta$. The error decreases by roughly one order of magnitude per decade as $\delta$ goes from $10^{-2}$ to $10^{-6}$, indicating fast practical convergence.
>
> | $\delta$ | 1e-2 | 1e-3 | 1e-4 | 1e-5 | 1e-6 |
> |---|---:|---:|---:|---:|---:|
> | Gradient Accuracy | 3.00e-4 ± 9.50e-5 | 4.24e-5 ± 1.84e-5 | 2.72e-6 ± 3.91e-7 | 3.63e-7 ± 1.37e-7 | 8.55e-8 ± 3.57e-8 |

---

> > ### Author Rebuttal · Reviewer_a7UL · 2026-04-02
> >
> > All my concerns are resolved. I will maintain a positive evaluation of the paper. But as I am not super familiar with the settings where such techniques are needed and how interesting they are, I will keep my score of WA, but it should be understood as leaning more towards accept.

---

> > > ### Author Response · Authors · 2026-04-05
> > >
> > > We sincerely thank you for your careful reading of our paper. We are grateful that you found your concerns resolved and that your evaluation leans towards acceptance. If you have any remaining questions about the practical applications or the broader significance of differentiable optimization, we would be happy to provide further clarification.

---

### Decision · Program_Chairs · 2026-04-30

**Decision:**

Accept (regular)

**Comment:**

The reviewers were at a consensus in favor of accepting this submission. I also agree with this assessment. The particular strengths of the paper includes:
- a simple but original idea for an important problem,
- the soundness of the results, with an effective reduction in the dimension of the linear system required during the backward pass,
- comprehensive and convincing numerical experiments comparing against state-of-the-art approaches, demonstrating scalability and speedup advantages.

The main weakness was the lack of sensitivity analysis for certain parameters, and the authors have addressed this in the rebuttal phase. The reviewers have also listed some minor presentation issues to be fixed. I recommend the authors to incorporate these changes in the final revision.